# RISC-interacting clearing 3'- 5' exoribonucleases (RICEs) degrade uridylated cleavage fragments to maintain functional RISC in *Arabidopsis thaliana*

Zhonghui Zhang[1,2†], Fuqu Hu[1,2†], Min Woo Sung[1], Chang Shu[1], Claudia Castillo-González[1,2], Hisashi Koiwa[3], Guiliang Tang[4], Martin Dickman[2], Pingwei Li[1*], Xiuren Zhang[1,2*]

[1]Department of Biochemistry and Biophysics, Texas A&M University, College Station, United States; [2]Institute for Plant Genomics and Biotechnology, Texas A&M University, College Station, United States; [3]Department of Horticulture, Texas A&M University, College Station, United States; [4]Department of Biological Sciences, Michigan Technological University, Houghton, United States

*For correspondence: pingwei@ tamu.edu (PL); xiuren.zhang@ tamu.edu (XZ)

†These authors contributed equally to this work

Competing interests: The authors declare that no competing interests exist.

**Abstract** RNA-induced silencing complex (RISC) is composed of miRNAs and AGO proteins. AGOs use miRNAs as guides to slice target mRNAs to produce truncated 5' and 3' RNA fragments. The 5' cleaved RNA fragments are marked with uridylation for degradation. Here, we identified novel cofactors of Arabidopsis AGOs, named RICE1 and RICE2. RICE proteins specifically degraded single-strand (ss) RNAs in vitro; but neither miRNAs nor miRNA*s in vivo. RICE1 exhibited a DnaQ-like exonuclease fold and formed a homohexamer with the active sites located at the interfaces between RICE1 subunits. Notably, ectopic expression of catalytically-inactive RICE1 not only significantly reduced miRNA levels; but also increased 5' cleavage RISC fragments with extended uridine tails. We conclude that RICEs act to degrade uridylated 5' products of AGO cleavage to maintain functional RISC. Our study also suggests a possible link between decay of cleaved target mRNAs and miRNA stability in RISC.

## Introduction

RNA silencing is a fundamental mechanism for regulating gene expression in diverse biological contexts in eukaryotes. The key silencing effector is a ribonucleoprotein complex (RISC) that is composed of AGOs and small RNAs (sRNAs). sRNAs, including small interfering RNAs (siRNAs) and microRNAs (miRNAs), are produced by Dicers or Dicer-like enzymes as duplexes from longer fully-complementary double-stranded (ds) RNAs, or from ds RNAs with imperfectly-folded hairpin structures (*Achkar et al., 2016*; *Kim et al., 2009*; *Sanei and Chen, 2015*). Subsequently, the sRNA duplexes are loaded into AGO proteins; and one strand (siRNA passenger strand or miRNA*) is ejected whereas the other (siRNA guide strand or miRNA) is retained in RISC. By using the retained sRNAs as guides, AGO proteins specify the correct target mRNAs and repress gene expression (*Chandradoss et al., 2015*; *Salomon et al., 2015*). For targets with partial complementarity, silencing is initially primed through translational repression and followed by mRNA destabilization (*Bazzini et al., 2012*; *Djuranovic et al., 2012*). Degradation of such an mRNA target is accomplished

**eLife digest** DNA contains all the information needed to build a body, yet molecules of RNA carry these instructions to the sites in the cell where they can be used. Cells must control how much RNA they produce in order to ensure that they develop properly and can respond well to their environment. RNA silencing refers to a collection of mechanisms that use smaller RNA molecules called microRNAs to incapacitate certain RNA molecules and selectively switch off the genes that encode them to stop more from being made.

One key player in RNA silencing is the multi-protein complex called RISC, which contains microRNA and a group of proteins called AGOs. Once the microRNA has identified its RNA target, the AGOs cut the RNA into two pieces, known as the 5' cleavage fragment and 3' cleavage fragment. The two resulting fragments need to be cleared away swiftly, so that the RISC can move on to the next target. While it was known how the 3' cleavage fragment was removed, it was less clear how the 5' cleavage fragment was dealt with.

Previous studies had shown that the 5' cleavage fragment was marked with a chemical called uridine, which somehow signals to the RISC that this fragment needs to be destroyed. Now, using biochemical techniques, Zhang et al. have identified two new proteins in the model plant Arabidopsis that attach to the AGO proteins and degrade the 5' cleavage fragments that are marked with uridine. The two proteins are named RICE1 and RICE2.

Zhang et al. then analyzed the three-dimensional shape of RICE1 and identified the 'active' region that is responsible for degrading the RNA fragments. When these active regions were blocked, the microRNA levels were low, but the uridine-marked 5' cleavage fragments were high. Also, the RISC complex could not work properly, which lead to problems during the development of the plant. These results suggest that RICE proteins degrade 5' cleavage fragments modified with uridine to activate RISC.

RICE proteins are conserved between plants and animals, and it is likely that their counterparts in humans will have a similar role to the plant proteins. The next challenge will be to explore how RICE proteins work in more details, which may lead to new ways to manipulate the levels of microRNAs to change the architecture of the plant and to improve their tolerance to various stress conditions.

by a combination of deadenylation, decapping, and subsequently by 5'-to-3' exonucleolytic decay (*Jonas and Izaurralde, 2015*).

For highly complementary targets, RISC activity generally results in 5' and 3' cleavage fragments (*Bartel, 2009*). Both 5' and 3 cleavage products undergo various decay processes. In mammals, the 3' cleavage fragments are canonically degraded in the 5' to 3' direction by an exonuclease 1 (XRN1). In Arabidopsis, the 3' fragment is accumulated at a higher level in loss-of-function mutant of *XRN4* compared with the amount in wild type (*Souret et al., 2004*). Since Arabidopsis *XRN4* is an ortholog of mammalian *XRN1*, the 5'-to-3' decay of RISC 3' cleavage fragments appears to be implemented through an evolutionarily conserved mechanism.

RISC 5' cleavage fragments turn over rapidly reflected by the relative inability to detect them when compared with 3' cleavage fragments. However, the RNA decay mechanism for 5' cleavage fragments has remained unclear. Previous studies showed that 5' cleavage fragments from miRNA-RISC activity are typically uridylated at their 3' ends throughout eukaryotes; and that this nontemplated modification is a signature license for immediate decay (*Ren et al., 2014*; *Shen and Goodman, 2004*). In human cells, the uridylation of the RISC 5' cleavage products is fulfilled through the terminal uridylyl transferases (TUTs) (*Lim et al., 2014*). The uridylation promotes decapping of RISC 5' products, and subsequently 5'-to-3' exonucleolytic decay by XRN1 (*Song and Kiledjian, 2007*). In Arabidopsis, HEN1 suppressor 1 (HESO1) uridylates the 5' fragments and stimulates their degradation, since a few 5'-cleavage fragments display modest overaccumulation in *heso1* mutants (*Ren et al., 2014*). Notably, HESO1 was initially recovered as a miRNA nucleotidyl transferase. HESO1 functions together with UTP:RNA uridylyl transferase one to promote miRNA degradation in absence of canonical miRNA methylation (*Ren et al., 2012*; *Tu et al., 2015*; *Wang et al., 2015*). In Arabidopsis, different pathways might account for RNA decay of RISC 5' cleavage fragments. It has

been shown that 5' cleavage fragments accumulate in *xrn4* mutant in Arabidopsis; and obviously XRN4 catalyzes 5'-to-3' degradation of the fragments in a way similar to clearing RISC 3' fragments. The RNA exosome also appears to contribute to degrade the 5' cleavage fragments because their abundance is increased in the loss-of-function mutant of *SKI2/3/8*, Arabidopsis orthologs of RNA exosome subunits in yeast (*Branscheid et al., 2015*). In contrast, accumulation of RISC 5' fragments is not enhanced in the mutants of additional subunits and even an *Rrp6* ortholog, a core 3'-to-5' exonuclease in the RNA exosome (*Branscheid et al., 2015*). Therefore, whether these pathways represent the totality of mechanisms for degradation of uridylated 5' cleavage fragments remains elusive.

miRNA targets not only serve as substrates for RISC activity, but also influence RISC function and miRNA stability. A pioneering study in plants shows that target mimicry can act as an endogenous decoy for miRNAs, resulting in unproductive RISC and miRNA destabilization (*Franco-Zorrilla et al., 2007*). Similar phenomena including miRNA sponges and competing endogenous mRNA (ceRNAs) that contain multiple miRNA-binding sites can modulate RISC activity and effectively inhibit miRNA function in animal systems (*Ebert and Sharp, 2010*; *Salmena et al., 2011*; *Rubio-Somoza et al., 2011*). In these organisms, miRNAs recognize target mRNAs through seed pairing (*Bartel, 2009*). Extensive pairing of 3' miRNAs to target RNAs triggers miRNA trimming and tailing and an accompanying loss of mature miRNAs (*Ameres et al., 2010*; *Xie et al., 2012*). In human cells, highly complementary target RNAs destabilize the RISC and accelerate release of the guide RNA from AGO2 whereas partially complementary targets attenuate unloading of sRNAs and increase their stability (*De et al., 2013*). Due to the presence of prevalent mismatches between the 3' end of a guide RNA and its target in mammals, the majority of identified miRNA targets do not destabilize the interaction (*Bartel, 2009*). In contrast, plant miRNAs are nearly perfectly complementary to their target RNAs; and miRNA-RISC canonically functions to cleave target RNAs despite coherent presence of translation repression (*Li et al., 2013*). However, whether RISC cleavage products regulate RISC function and miRNA abundance is unknown.

Arabidopsis encodes nine functional AGOs, among which, AGO1 is a principal contributor to RNA silencing as it recruits most miRNAs, and a variety of siRNAs (*Mi et al., 2008*; *Wang et al., 2011*). AGO10 is the closest genetic paralog of AGO1, but functions to specifically sequester a group of miRNAs, miR165/166, to antagonize their silencing activity through AGO1 (*Zhu et al., 2011*; *Zhou et al., 2015*; *Yu et al., 2017*). Here, we successfully used proteomics analysis to identify a novel AGO10-bound partner, RICE1. We showed that RICE1 and its genetic paralog, RICE2, interacted with both AGO10 and AGO1, suggesting a common role in regulation of miRNA-RISC activity. RICE1 and RICE2 function as 3'-to-5' exoribonucleases that specifically degraded ss RNAs. We found that overexpression of *RICE*s increased miRNA levels, whereas downregulation of *RICE1* and *RICE2* through artificial miRNA technology decreased miRNA accumulation. We also determined the crystal structure of RICE1 and observed that it formed a homohexameric ring with the active sites uniquely embedded at the interface between monomers. We further pinpointed the critical residues required for RICE1 catalytic activity and oligomerization. Whereas oligomerization-defective RICE1 variants were unstable in vivo, introduction of catalytically-inactive RICE1 significantly reduced miRNA levels in vivo, reminiscent of loss-of-function mutants of *rice1 rice2*. Importantly, catalytically inactive RICE1 also caused the accumulation of longer uridylated 5' cleavage RNA fragments from RISC activity. We propose that RICE1 and RICE2 act to clear 5' uridylated fragments from RISC cleavage and to facilitate RISC recycling. Moreover, our study also suggests that the turnover of miRNA targets and miRNA accumulation might be related in vivo.

## Results

### Identification of RICEs as AGO10-bound cofactors from *Arabidopsis thaliana (A. thaliana)*

To identify AGO10 cofactors, we generated stable *A. thaliana* transgenic suspension cell lines over-expressing *Flag-4Myc-AGO10* to mimic plant stem cells where AGO10 is specifically expressed (*Figure 1A,B*). Then, we purified the AGO10 complex through two-step immunoprecipitation and resolved the complexes on SDS-PAGE gradient gels (*Figure 1C*). Distinct bands that were absent from the control IP with non-transgenic suspension cells were analyzed by mass spectrometry (MS) (*Figure 1C*). We identified numerous cofactors, including HSP90 and cyclophilins, which are known

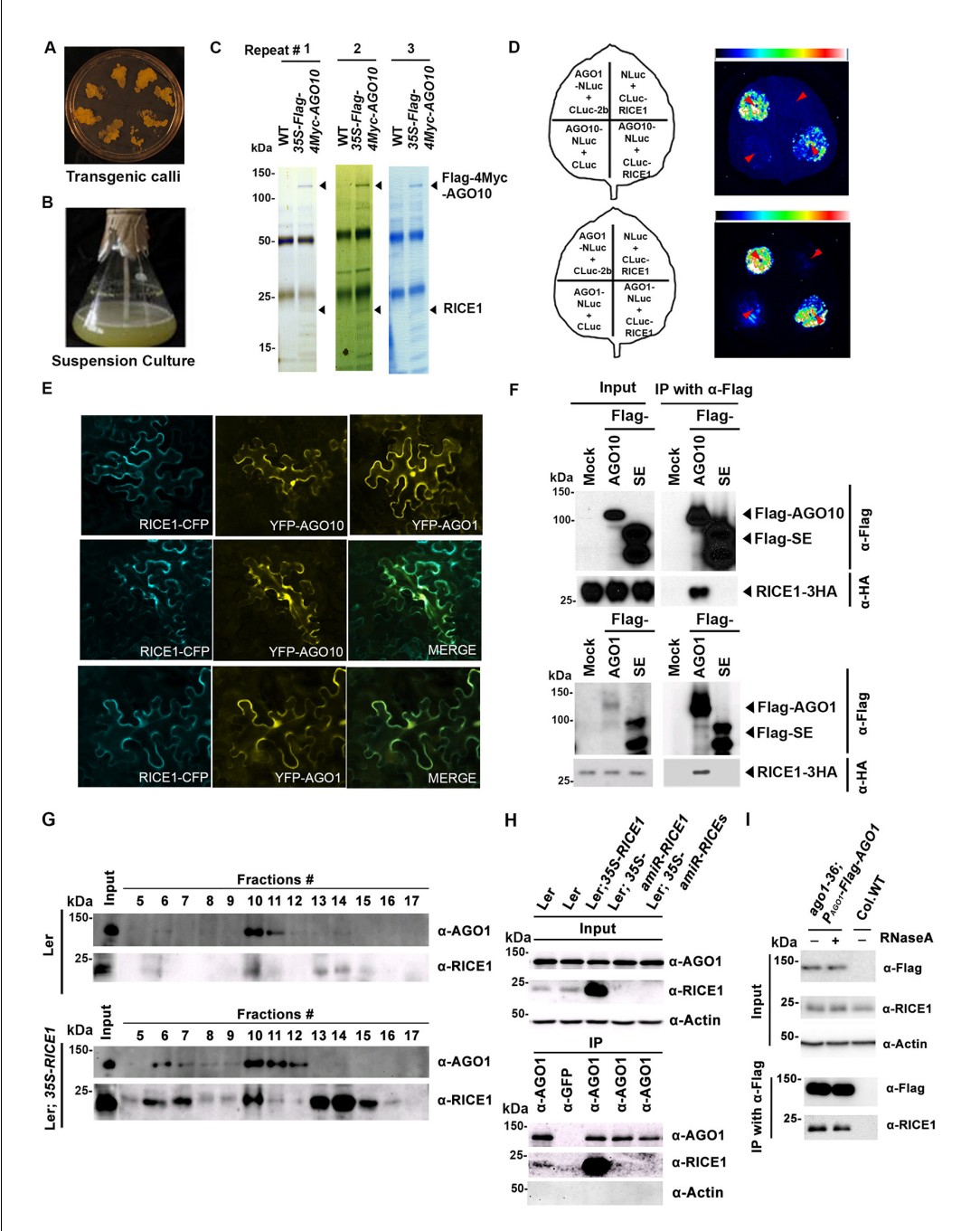

**Figure 1.** RICE1 is a novel RISC-bound cofactor in Arabidopsis. (**A** and **B**) Generation of transgenic calli and cell lines expressing *35S-Flag-4Myc-AGO10*. (**C**) Gel-code blue stained SDS-PAGE of purified AGO10 complexes for proteomics analysis. (**D–F**) Specific RICE1-AGO interaction was confirmed in *N. benthamiana (N. bentha.)* by LCI (**D**), co-localization (**E**), and co-IP (**F**) assays. In (**D**), the schemes of leaves show different combinations of infiltrated constructs that were fused either to N-terminal (NLuc) and C-terminal (CLuc) regions of luciferase. The LCI photographs on right showed the signals resulting from the protein-protein interaction. The red arrows indicate the infiltration positions. A combination of AGO1 and CMV 2b (*Zhang et al., 2006b*) serves as a positive control. In (**E**), RICE1-CFP and YFP-AGO proteins, when expressed alone in the *N. bentha*, were shown in top panels. The RICE1-CFP and YFP-AGOs, when co-expressed and detected in CFP and YFP channels separately, were shown in middle and bottom panels, respectively. Note: co-localization of RICE1-CFP and YFP-AGOs was observed predominantly in cytoplasm. In (**F**), IP was done with an anti-Flag antibody. Western blot analyses were conducted using anti-Flag, or -HA antibodies. SE is a negative control. (**G–I**) Specific interaction between endogenous RICE1 and AGO1 in Arabidopsis was confirmed by SEC (**G**) and co-IP (**H** and **I**) assays. IP was performed with anti-AGO1 (**H**) or anti-Flag (**I**) and western blot analyses were detected with anti-AGO1 (or -Flag), -RICE1 and -actin antibodies. Actin is a negative control. In (**I**), the crude extract was applied with (+) or without not (−) 50 µg/ml RNase A before co-IP.

*Figure 1 continued*

The following figure supplement is available for figure 1:

**Figure supplement 1.** Identification and experimental confirmation of RICEs as novel Arabidopsis RISC-bound cofactors.

to be required for RISC assembly (*Iki et al., 2010*, *2012*), as well as novel factors (*Figure 1C*; *Figure 1—figure supplement 1A*). The results of the AGO10 complex isolation and MS analysis were reproducible in three independent experiments (*Figure 1C*). Here, we focus on *RICE1* (*At3g11770*), which is annotated as a 23 kDa polynucleotidyl transferase and ribonuclease H-like superfamily protein in the TAIR database. *RICE1* has a close paralog (*At5g06450*, *RICE2*) that was previously known as *AtDECP*, Arabidopsis DnaQ-like 3′-to-5′ exonuclease domain-containing protein (*Perry et al., 2006*; *Smith et al., 2013*). RICE1 and RICE2 share 67% sequence identity, but their functions have not been characterized.

To examine whether RICE1 is a *bona fide* partner of AGO10, we first carried out a split luciferase complementation assay (LCI) (*Zhang et al., 2011*). In our LCI assays, both AGO10 and AGO1 displayed LUC complementation with RICE1 (*Figure 1D*), as did AGO1 with the positive control, *Cucumber mosaic virus*-encoded 2b protein (CMV 2b) (*Zhang et al., 2006b*), suggesting that RICE1 may be a common factor in various RISCs (*Figure 1D*). Next, using confocal microscopy, we observed that both AGO1 and AGO10 predominantly co-localized with RICE1 in the cytoplasm, suggesting a potential function of RICE1 in processing nucleic acids in the cytoplasm (*Figure 1E*; *Figure 1—figure supplement 1B*). To further investigate RICE-AGO interactions, we performed co-immunoprecipitation (co-IP) experiments. To this end, we transiently expressed RICE1-3HA with either Flag-AGOs or -Serrate (SE), a control protein involved in the miRNA pathway (*Manavella et al., 2012*), by agroinfiltration in *Nicotiana benthamiana* (*N. bentha*). We conducted an IP with anti-Flag antibody and detected co-immunoprecipitates with anti-HA antibody. RICE1-3HA was indeed co-immunoprecipitated with both AGO1 and AGO10, but not with the control protein (*Figure 1F*). We next conducted size exclusion chromatography (SEC) assays of Arabidopsis extracts in both wild type and *35S-RICE1* transgenic plants. Western blot with antibodies against endogenous AGO1 and RICE1, respectively, indicated that a fraction of RICE1 protein was co-eluted with AGO1 (*Figure 1G*). Additional IP experiments with the same extracts demonstrated the consistent coexistence of the two proteins in vivo (*Figure 1H*). Similar co-IP results were also obtained in $P_{AGO1}$-Flag-AGO1 complementation line. Furthermore, RNase A treatment in the co-IP process did not abolish the interaction between RICE1 and AGO1, indicating that their interactions were RNA-independent (*Figure 1I*). Finally, in vitro pull-down assays showed that 6His-SUMO-RICE1 and -RICE2, but not 6His-SUMO, specifically and directly interacted with maltose-binding protein (MBP)-AGO1 and -AGO10, but not control protein MBP, (*Figure 1—figure supplement 1D–F*, lanes #1, 9 and 10). Moreover, RICEs interacted with several fragments of AGO10 proteins – AGO10 NC (127-336), AGO10 PAZ (337-474), and AGO10 PIWI (625-988) – with an apparently stronger affinity to the PAZ domain (*Figure 1—figure supplement 1F*, lanes # 3–8). In contrast, neither of the variable parts of AGO10 or AGO1 N-terminal domains could be pulled down by RICE proteins (*Figure 1—figure supplement 1F*, lanes #2 and 11). Collectively, these experiments demonstrated that RICEs are indeed novel AGO1/10-interacting proteins in plants.

## RICE1 and RICE 2 act as 3′-to-5′ exoribonucleases specifically targeting single-stranded RNAs

Computational analysis predicted that RICEs might function as RNase H proteins (from TAIR website) or DnaQ-like nucleases (*Smith et al., 2013*). To characterize the biochemical function of RICEs, we prepared RICE proteins from *E. coli* through Ni-NTA affinity purification followed by SEC (*Figure 2A*; *Figure 2—figure supplement 1*). The major peaks in the SEC profiles of RICE1 and RICE2 corresponded to molecular weights of 130 and 180kD, respectively (*Figure 2—figure supplement 1B,C*); that are approximately six times the monomer size, indicating that RICE1 and RICE2 most likely form oligomers in solution. To identify *bona fide* substrates for RICE1, we tested RICE1 enzymatic activity with 5′ $^{32}$P-labelled ss 21-nucleotide (nt) RNA or DNA using various reaction

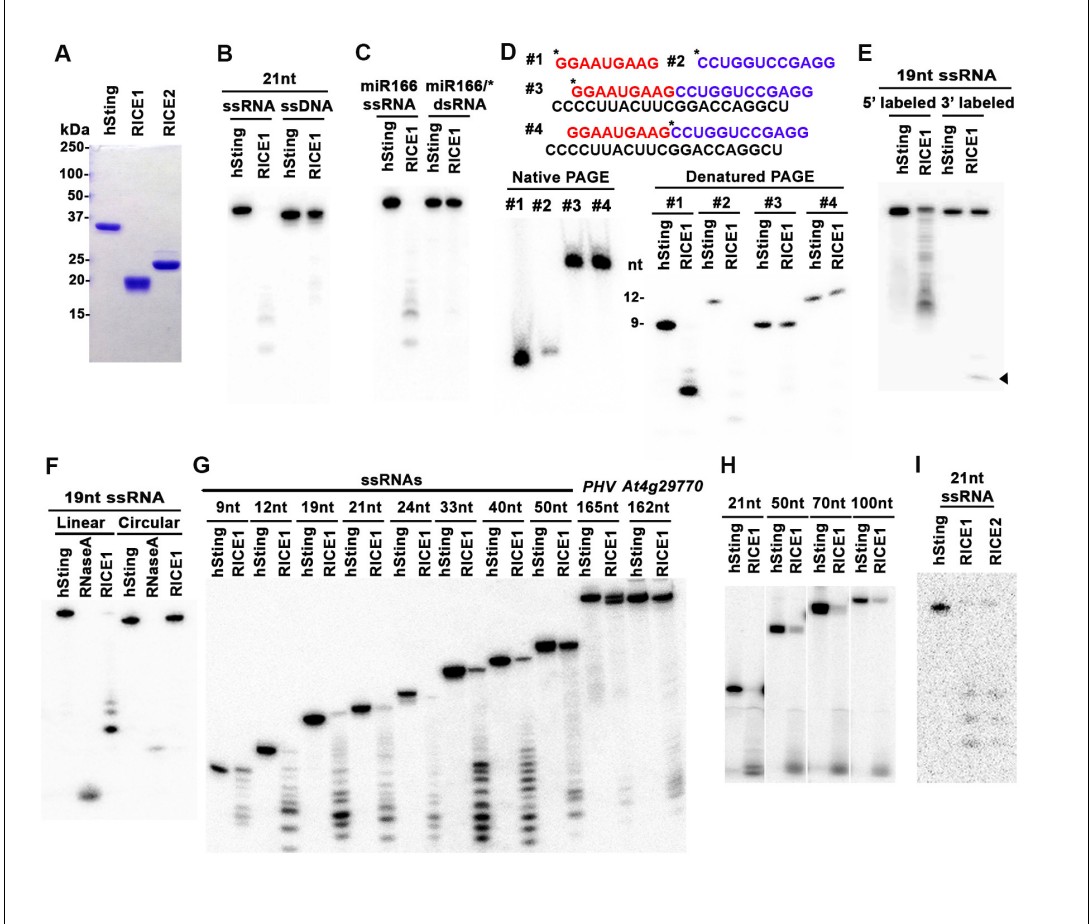

**Figure 2.** RICEs are 3'-to-5' exoribonucleases preferring ss RNA. (**A**) Coomassie brilliant blue staining of purified RICEs. hSting protein serves as a control throughout experiments. (**B**) RICE1 favored ss RNA over ss DNA as a substrate. (**C**) RICE1 did not degrade ds RNA. miR166/*, the $^{32}$P-labeled miR166 was annealed with unlabeled miR166*. (**D**) RICE1 did not degrade nicked but not released miRNA/*. Top panel shows positions of labeled nucleotides (*) and schematic diagram of nicked but still annealed miRNA/*. The nucleotides in red are the 5' 9-nt fragment of miR166* and the nucleotides in blue are the 3' 12-nt fragment of miR166*. (**E**) RICE1 was a 3'-to-5' ribonuclease. Released single nucleotide is shown with an arrowhead. (**F**) RICE1 was an exoribonuclease. The circular substrates were generated by the self-ligation of 5' labeled linear RNAs with T4 RNA ligase. (**G**) RICE1 efficiently degraded ss sRNA but not long RNA with secondary structures. Partial mRNAs of *PHV* and *At4g29770* (165 nt and 162 nt in length) serves as long ss RNA substrates. (**H**) RICE1 efficiently degraded both sRNA and long RNA homopolymer (poly-As in 21-, 50-, 70-, and 100-nt length). (**I**) RICE2, similar to RICE1, is a 3'-to-5' exoribonuclease. All the substrates in the enzymatic assay were 5' $^{32}$P-labeled except the ones in (**E**) as indicated.

The following figure supplement is available for figure 2:

**Figure supplement 1.** Purification of RICE1 and RICE2 proteins.

conditions, including the one described for its human structural homolog, Werner Syndrome Helicase exonuclease (WRN-exo) (*Perry et al., 2006*). Our results show that RICE1, in contrast to the control protein, human Sting protein (hSting) (*Shu et al., 2012*), which underwent the same purification process, degraded the ss RNA, but not ss DNA, into shorter fragments, indicating that RICE1 principally acts as an RNase and not a DNase (*Figure 2B*). Next, we performed cleavage assays of RICE1 on ds RNA substrates. To make the ds RNA substrate, miR166 was 5' labeled and annealed to cold miR166*. Enzymatic assays showed that RICE1 readily degraded miR166 but not the miR166/* duplex, indicating that RICE1, unlike the worm exonuclease Eri-1 that degrades siRNA duplexes (*Kennedy et al., 2004*), specifically degrades ss RNAs (*Figure 2C*).

In humans and fungi, following the loading of siRNA duplexes, the passenger strand is nicked by catalytically-active AGO proteins and is subsequently cleared via nucleases such as C3PO and QIP

(*Liu et al., 2009*; *Maiti et al., 2007*; *Ye et al., 2011*). To investigate whether RICE1 function resembles C3PO or QIP and removes the nicked passenger strand from sRNA duplexes, a 5' $^{32}$P labeled 12-nt RNA fragment and cold 9-nt RNA fragment fully complementary to miR166 were annealed to cold miR166a (*Figure 2D*) to generate a nicked sRNA duplex. Similarly, an analogous nicked duplex was prepared by annealing 5' $^{32}$P labeled 9-nt RNA fragment with cold 12-nt fragment and miR166a (*Figure 2D*). While RICE1 cleaved the 5' labeled 9-nt and 12-nt ss sRNAs effectively, the 9-nt and 12-nt RNA fragments in the duplex forms were resistant to RICE1 degradation (*Figure 2D*). Thus, RICE1, unlike previously characterized C3PO and QIP, did not degrade unreleased nicked ss RNAs.

To distinguish whether RICE1 is a 5'-to-3' or 3'-to-5' ribonuclease, we repeated the RNase assays with the RNA substrates labeled on different ends. When a 5' labeled substrate was used for the assay, truncated fragments of various sizes were observed; whereas when a 3' labeled ss RNA substrate was used, only a single one-nucleotide cleavage product was observed. This RNA substrate was more stable than the 5' labeled RNA and we infer that the 3' labeled phosphate inhibited the target degradation (*Figure 2E*). These results indicate that RICE1 is a 3'-to-5' ribonuclease that cleaves nucleotides off the RNA substrates progressively from the 3' terminus. To further determine whether RICE1 is an exo- or endo-ribonuclease, a 5'-labeled 21nt ss RNA was circularized by T4 RNA ligase. While the circularized ss RNA was effectively degraded by RNase A, this RNA was resistant to RICE1 activity, suggesting that RICE1 is an exoribonuclease (*Figure 2F*).

We next tested the length requirement of RNA substrates for RICE1. ss RNAs, with lengths ranging from 9 to 165-nt, were 5' $^{32}$P labeled and subjected to RICE1 treatment. We observed that ss RNAs with lengths less than 40-nt were readily degraded by RICE1 while the RNAs that were 50-nt or longer were less vulnerable to the exonucleolytic activity of RICE1 (*Figure 2G*). However, considering that double-strand secondary structures are more likely to occur in the long ss RNAs than short ones, we tested RICE1 enzyme activity against RNA homopolymers of different lengths. Consequently, we noticed that RICE1 enzyme activities on these 21-nt to 100-nt substrates were comparable (*Figure 2H*), suggesting that RICE1 favored both small ss RNAs and long ss RNAs without secondary structure. We repeated the nuclease activity assay with RICE2 and observed that RICE2 functioned essentially identical to RICE1 (*Figure 2I*; data not shown). Taken together, both RICE1 and RICE2 are exoribonucleases specifically targeting ss RNAs.

## RICE1 and RICE2 positively regulate miRNA accumulation in vivo

To study the in vivo function of *RICE1* and *RICE2* genes, we examined their expression profiles using transgenic plants expressing a *GUS* reporter gene under the native *RICE* promoters. *RICE1* promoter was ubiquitously active in various tissues and organs throughout development including germinating seeds, cotyledons, leaves and roots of young seedlings and adult plants, stems and inflorescence (*Figure 3—figure supplement 1*). In contrast, *RICE2* promoter activity was more restricted to specific niches including shoot and root apical meristems, trichomes, and vascular veins, among other tissues (*Figure 3—figure supplement 1*). The partially overlapping expression domains of these two genes suggest their functional redundancy but also possible specificity for different biological processes.

What are the functions for RICE1 and RICE2 in vivo? Given that RICEs are housed in AGO1 and AGO10-centered complexes and that the proteins are 3'-to-5' exoribonucleases specifically degrading ss RNAs, our initial hypothesis was that RICEs might function to degrade AGO-bound miRNAs in a manner similar to SDN1 (*Ramachandran and Chen, 2008*). To test this, we examined miRNA accumulation in transgenic plants with altered levels of *RICE1* and *RICE2* in vivo. First, we generated transgenic plants expressing artificial miRNA constructs specifically targeting *RICE1* (*amiR-RICE1*), *RICE2* (*amiR-RICE2*), or both (*amiR-RICEs*) (*Figure 3A*; *Figure 3—figure supplement 2A*). Consistent with *GUS*-reporter line results, *RICE1* transcripts were much more abundant than *RICE2* transcripts (*Figure 3B*; *Figure 3—figure supplement 2B*). Analyses of sRNA and RNA blots showed that artificial miRNAs targeting individual or both genes were efficiently processed and correspondingly, levels of *RICE1* and/or *RICE2* transcripts decreased approximately ~70–90% compared to wild type plants (*Figure 3B*; *Figure 3—figure supplement 2A–B*). Knockdown of just *RICE1* or *RICE2* in the *amiR-RICE1* and *amiR-RICE2* transgenic lines seemed to have little effect on the levels of tested miRNAs (*Figure 3C–D*; *Figure 3—figure supplement 2C*). However, concurrent knockdown of both *RICE1* and *RICE2* consistently decreased the abundance of tested miRNAs relative to the control plants or the sibling lines with barely detectable artificial miRNAs (*Figure 3C–D*; *Figure 3—figure*

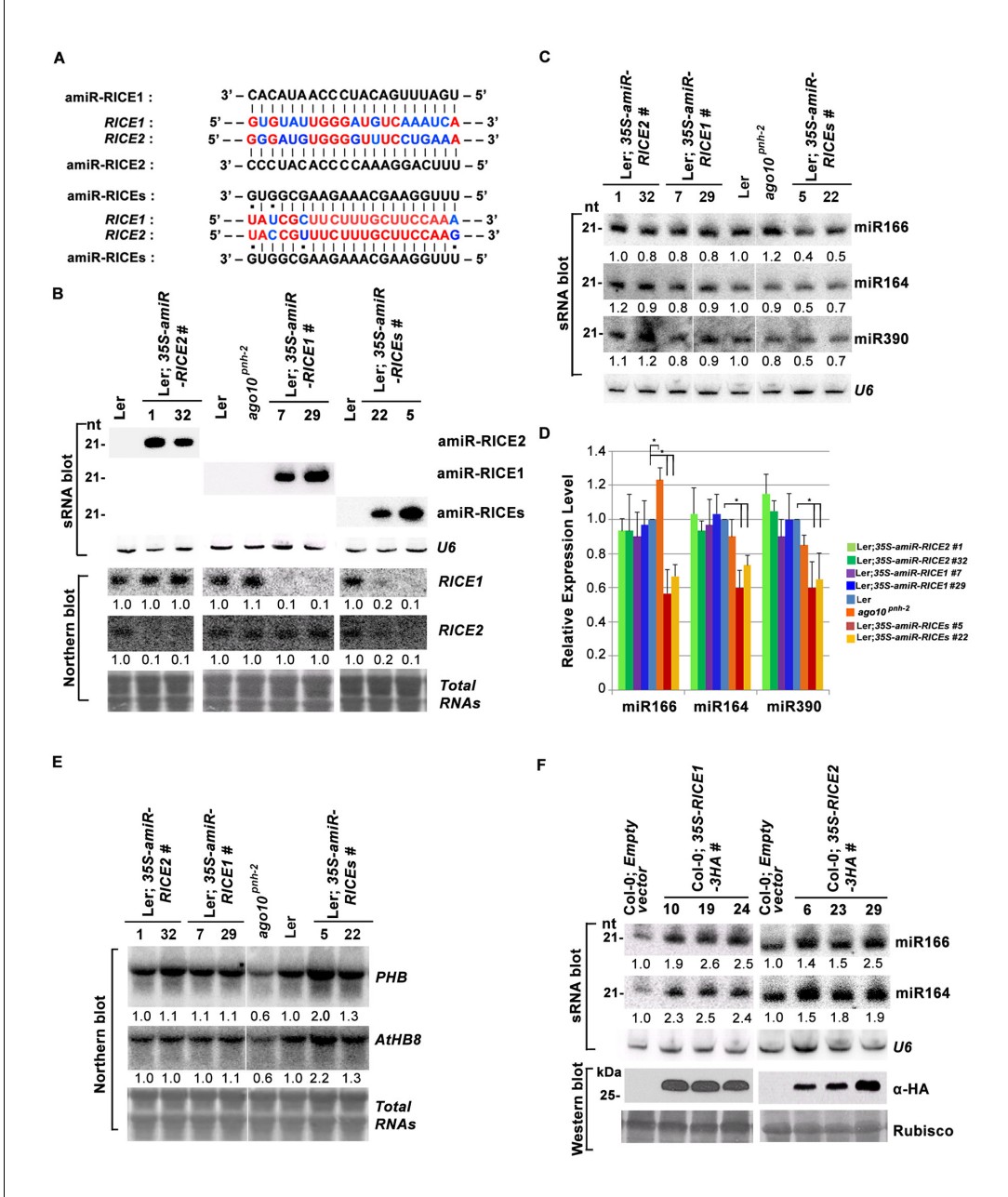

**Figure 3.** RICE1 and RICE2 are positive regulators for miRNA accumulation in vivo. (**A**) Sequence alignment of artificial miRNAs and their targets. The nucleotides in red are perfect matched region between *RICE1* and *RICE2* target sites, while the nucleotides in blue are mismatched region between *RICE1* and *RICE2* target sites. (**B**) Generation of artificial miRNA transgenic lines targeting specific *RICE1*, *RICE2*, or both transcripts. (**C–E**) Concurrent knockdown of *RICE1* and *RICE2* consistently reduced miRNA abundance (**C** and **D**) and upregulated the targeted transcripts (**E**). (**F**) Ectopic expression of RICE1-3HA in wild-type Ler significantly enhanced accumulation of tested miRNAs. sRNA and RNA gel blot analyses were done with the indicated ³²P-labelled probes. *U6* and total RNAs stained with methylene blue serves as loading controls. For (**C**), additional two biological replicates were shown in ***Figure 3—figure supplement 2C***. Quantification of three biological replicates was statistically analyzed and the significant difference (Student t-test, p<0.05) shown by the asterisk (*) in (**D**). Also see in ***Supplementary file 1A***. Western blot analyses were performed with anti-HA antibody and Rubisco stained with Ponceau S serves a loading control. The amount of miRNAs or mRNAs in wild-type plants or transgenic plants harboring empty vectors was arbitrarily designated as 1.0, and the relative amount in the other lines was normalized to that of control plants.

The following figure supplements are available for figure 3:

**Figure supplement 1.** *RICE1* is ubiquitously expressed in various organs and tissues throughout different developmental stages whereas *RICE2* expression is more restricted and less abundant.

*Figure 3 continued*

**Figure supplement 2.** *RICEs* positively regulate miRNA accumulation in vivo.

*supplement 2C*). Correspondingly, mRNA transcripts targeted by the assayed miRNAs were upregulated (*Figure 3E*).

In parallel, we produced transgenic plants constitutively expressing RICE1 and RICE2 tagged with 3 HA at the C-termini. SEC and in vitro enzymatic assays showed that purified HA tagged RICE1 proteins had biochemical features identical to wild type RICE1 protein, suggesting that these tagged proteins are able to recapitulate functions of RICE1 in vivo (*Figure 3—figure supplement 2D–F*). RNA blot analysis showed that overexpression of RICE1 or RICE2 increased the levels of all tested miRNAs compared to the control plants transformed with empty vectors or sibling plants with undetectable accumulation of tagged RICE1 and RICE2 (*Figure 3F*). We repeated these experiments with transgenic plants expressing CFP tagged RICEs in the wild type or *ago10$^{pnh-2}$* mutant background and obtained similar results (*Figure 3—figure supplement 2G,H*). Given that the accumulation of free miRNAs outside RISC in Arabidopsis is negligible, the levels of miRNAs detected here are proposed to represent the accumulation of miRNAs in the mature RISC in vivo (*Wang et al., 2011*). Thus, RICEs, different from SDN1, function as positive regulators for miRNA accumulation in RISC.

## RICE1 exhibits a DnaQ-like exonuclease fold and forms a homohexamer with altered active site structure

To elucidate the catalytic mechanism of RICE1, we have determined the crystal structure of RICE1 at 2.9 Å resolution (*Supplementary file 2*). RICE1 crystallizes in space group I4$_1$22 with three molecules in the asymmetric unit. RICE1 molecules form two donut-shaped hexamers that interact with each other in a face-to-face manner in the crystal lattice (*Figure 4—figure supplement 1A*). A similar hexamer was also observed in the crystal structure of RICE2 crystallized in a different unit cell (*Smith et al., 2013*). RICE1 exhibits an α/β fold with a central seven-stranded β-sheet sandwiched between eight α-helices (*Figure 4A,B*). Structural analysis of RICE1 using the DALI server (*Holm and Rosenström, 2010*) revealed that RICE1 exhibits structural homology to the DnaQ family of 3'-to-5' exonucleases including WRN-exo (r.m.s.d 2.4 Å to 2E61) (*Perry et al., 2006*) and recently reported RICE2 (r.m.s.d 0.7 ~ 1.2 Å) (*Smith et al., 2013*) (*Figure 4—figure supplement 1B,C*). RICE1 also shows considerable structural similarity to ribonuclease D (r.m.s.d 2.7 Å to 1YT3) (*Zuo et al., 2005*).

The DnaQ-like exonuclease superfamily members have four conserved acidic residues (DEDD) in their active sites (*Perry et al., 2006*). The well-conserved WRN-exo active site is located within a large cavity on one face of the molecule. A comparison of RICE1 with WRN-exo and other members of the DnaQ family revealed that the DEDD motif is not conserved in the active site of RICE1 (*Figure 4—figure supplement 1B*). Instead, the glutamate and the last aspartate residues in the DEDD motif are replaced by Tyr54 and Glu187 (*Figure 4A,C–E*). In addition, the highly conserved tyrosine residue (Tyr212) near the active site of WRN-exo is replaced by Ala183 in RICE1. To confirm that these residues form the active site of RICE1, we expressed three RICE1 mutants, D52A, Y54S and E187A, and analyzed their oligomerization and catalytic activity. The SEC profiles of these mutants were identical to that of the wild type enzyme, indicating that these mutations do not affect the oligomerization of RICE1 (*Figure 5A*). However, enzymatic activity assays showed that these mutations eliminated or severely reduced the exoribonuclease activity of RICE1 (*Figure 5B–C*), indicating that these residues indeed form the active site of RICE1 and are indispensable for its exonuclease activity.

Another structural feature of the DnaQ-like exonuclease is the involvement of two metal ions at the active sites (*Perry et al., 2006*; *Zuo and Deutscher, 2001*). Notably, no obvious electron density corresponding to the active site Mg$^{2+}$ ion of the DnaQ-like nucleases was observed near the active site of RICE1 (*Figure 4E*), despite the inclusion of 5 mM of MgSO$_4$ in the protein sample for crystallization. Asp52 and Asp114, two residues that otherwise bind to divalent cations, interact with Lys102 and Arg127 of a neighboring RICE1 molecule. In line with this observation, EDTA did not affect the enzymatic activity of RICE1 (*Figure 4F*).

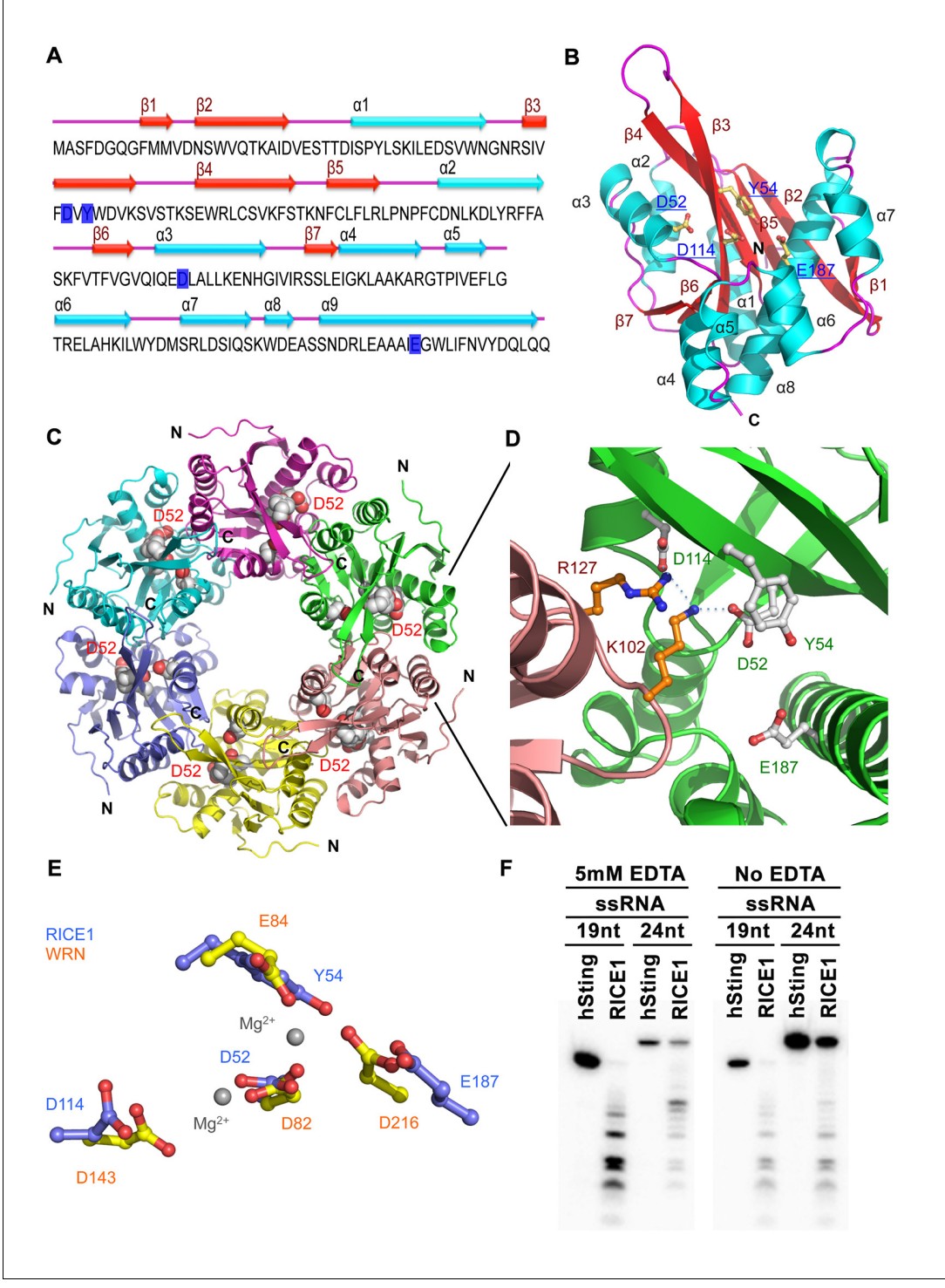

**Figure 4.** Crystal structure of RICE1. (**A**) The amino acid sequence and secondary structure of RICE1. (**B–D**) Structures of RICE1 monomer and hexamer. In (**B**) α-helices, β-strands, and loops are colored in cyan, green, and pink, respectively. In (**A** and **B**), residues (D52, Y54, D114 and E187) in the active site are highlighted in blue. In (**C**), each monomer is shown in different colors. Active site residues are shown by the sphere representations, with active site residues D52 of each monomer labeled. In (**D**), the active sites of RICE1 are located at the interface between RICE monomers. Key residues (D52, Y54, D114 and E187) in the active site are shown as ball-and-stick models. Salt bridges are denoted by blue dotted lines. (**E**) Overlay of active sites residues of RICE1 and WRN exonuclease. (**F**) Enzyme assays showed that EDTA did not affect the catalytic activity of RICE1.

*Figure 4 continued on next page*

*Figure 4 continued*

The following figure supplement is available for figure 4:

**Figure supplement 1.** Structural features of RICE1.

## Mutations in the catalytic site impair RICE1 function in vivo

To ask whether RICE1 catalytic activity is critical for the miRNA accumulation in vivo, we generated transgenic plants constitutively expressing catalytically-inactive RICE1 variants (D52A and Y54S). Although RICE1 D52A and Y54S mutations abolished the catalytic activity, they did not interfere with the RICE1-AGO10 interaction in the tobacco transient assay (data not shown). We obtained numerous positive transgenic plants expressing *35S-3HA-RICE1 D52A* and *35S-3HA-RICE1 Y54S* as determined by western blot analysis (*Figure 5—figure supplement 1A,B*). Overexpression of *35S-HA3-RICE1 D52A* and *35S-HA3-RICE1 Y54S* caused pleiotropic developmental defects including spoon-shaped cotyledons, twisted true leaves and infertile flowers (*Figure 5D*), phenocopying hypomorphic *ago1* mutants and transgenic plants expressing viral suppressors that disable the miRNA pathway in plants (*Zhang et al., 2006b*). Western blot analyses showed that the phenotypic severity was proportional to the expression levels of RICE1 mutants (*Figure 5E,G*). Together, all these results indicated that RICEs play important roles in plant growth and development.

Next, we measured miRNA expression in the transgenic plants. In a sharp contrast with RICE1 overexpression plants, over-accumulation of the catalytically-inactive forms of RICE1 D52A and RICE1 Y54S significantly decreased the levels of all tested miRNAs and trans-acting siRNA compared to Ler wild type and sibling transgenic plants with barely detectable transgene expression (*Figure 5E–G*; *Figure 5—figure supplement 1C,D*; Figure 7B). These results were reminiscent of the observation in *rice1 rice2* knockdown lines (*Figure 3C,D*; *Figure 5—figure supplement 1C*; *Figure 7—figure supplement 1D*), indicating that catalytically-inactive mutations act in a dominant negative manner on miRNA accumulation in vivo.

## Dimerization and oligomerization are essential for RICE1 function in vivo

Wild type RICE1 elutes predominantly as a hexamer in the SEC assay. Moreover, the active site of RICE1, unlike other DnaQ family members (*Perry et al., 2006*), is located at the interface between RICE1 subunits within the hexameric ring (*Figures 4C* and *6A*). Thus, the oligomerization of RICE1 seems critical for its catalytic activity. Structural analysis revealed extensive intermolecular interactions between RICE1 molecules within the hexameric ring. RICE1 molecules interact with each other mainly through electrostatic interactions (*Figure 6A–C*). Five salt bridges, between residues Arg47 and Asp166, Lys102 and Asp52, Lys102 and Asp114, Lys119 and Glu113, Arg127 and Asp114, are observed at the interface between RICE1 subunits (*Figure 6B*). Hydrogen bonds and van der Waals interactions make additional contributions to the interactions between RICE1 molecules. The total buried surface area between two neighboring RICE1 molecules is ~2800 Å$^2$. The two binding surfaces show both charge and shape complementarities to each other (*Figure 6B*). In summary, the structural analysis revealed that RICE1 exhibits a DnaQ family exonuclease fold with an altered active site that is located at the interface between RICE1 molecules.

To examine how oligomerization affects the properties of RICE1, we studied three RICE1 mutants in key residues at the RICE1 interface: R47E, K119E, and D114A. SEC showed that the K119E mutation did not affect RICE1 oligomerization (data not shown). By contrast, the R47E mutant displayed a distinct chromatographic profile compared to wild-type RICE1 with the major peak corresponding to the monomeric RICE1 (*Figure 6D*). Notably, the D114A mutant eluted as several peaks that correspond to the monomer, dimer and hexamer of RICE1 (*Figure 6E*). These results indicated that residues Arg47 and Asp114 are critical, for the oligomerization of RICE1. Importantly, nuclease assays showed that the enzymatic activity of the R47E mutant, but not the K119E mutant was abolished (*Figure 6F,G*; data not shown), indicating the oligomerization of RICE1 is required for the nuclease activity, in line with the observation that the active site of RICE1 is located at the interface between RICE1 subunits.

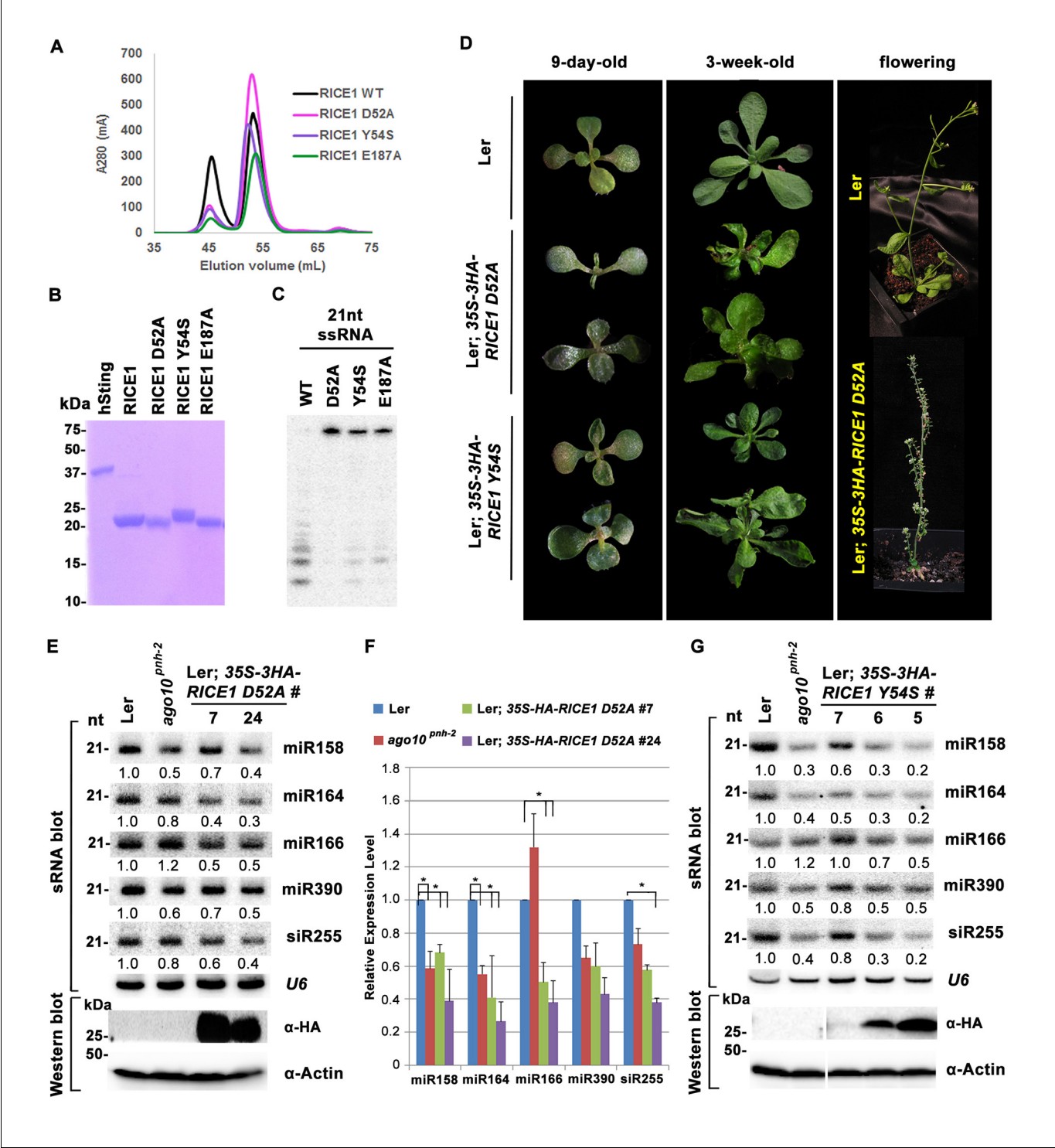

**Figure 5.** Mutations in catalytic sites impair RICE1 function in vivo. (**A**) Chromatograph of representative RICE1 variants with mutations in catalytic sites. (**B**) SDS-PAGE of purified RICE1 mutants and control protein hSting. (**C**) Representative RICE1 mutants were compromised exoribonucleases compared to wild type (WT). (**D**) Phenotypic anomaly of transgenic plants ectopically expressing catalytically-inactive RICE1 variants in different stages. Note: the severe sterility of *35S-3HA-RICE D52A* transgenic plants. (**E–G**) Ectopic expression of RICE1 variants caused dominant-negative effect on miRNA accumulation in vivo. In (**E**) and (**G**), sRNA blot analyses were done with the indicated $^{32}$P-labelled probes. The amount of miRNAs in wild-type plants was arbitrarily designated as 1.0, and the relative amount in the other lines was normalized to that of control plants. In (**F**), quantification of three biological replicates was statistically analyzed and the significant difference (Student t-test, p<0.05) shown by the asterisk (*). Additional biological

*Figure 5 continued on next page*

Zhang *et al*. eLife 2017;6:e24466. DOI: 10.7554/eLife.24466

*Figure 5 continued*

replicates were shown in **Figure 5—figure supplement 1C** and **Supplementary file 1B**. Western blot analyses were performed with anti-HA or anti-actin antibodies.

The following figure supplement is available for figure 5:

**Figure supplement 1.** Catalytic inactive RICE1 impairs the accumulation of miRNAs in vivo.

To examine how the oligomerization-compromised mutations affected the miRNA pathway, we also generated transgenic plants expressing *35S-3HA-RICE1 R47E* and *35S-3HA-RICE1 D114A*. Surprisingly, we seldom recovered positive transgenic plants by western blot screening of dozens of T1 transformants, although the transcript levels of these RICE1 variants were comparable with the ones extracted from the plants expressing *35S-3HA-RICE1 D52A/Y54S* (**Figure 6H–J**; data not shown). In line with this result, no obvious defects of development and miRNA accumulation were visible in any of RICE1 R47E and D114A transgenic lines (**Figure 6K**; data not shown). Together, these results indicated that disruption of protein oligomerization led to protein instability.

## RICE1 does not clear miRNA* during RISC assembly

Because RICEs function as positive regulators for miRNA accumulation in vivo, miRNAs themselves are unlikely the direct targets of RICEs. An alternative hypothesis is that RICEs might clear the miRNA* strand and facilitate incorporation of miRNA strands into AGO proteins. Because how miRNA/miRNA* is exactly loaded into AGO protein remains unclear in plants, we hypothesized that if RICEs removed miRNA* after the incorporation of miRNA/* into AGO proteins, then a defect in RICE function might lead to enrichment of miRNA* in AGO proteins. To test this, we examined the miRNA accumulation in AGO1 protein in the lines overexpressing the dominant negative mutant *RICE1 D52A*. The accumulation of AGO1 was slightly increased in the mutant, likely due to its feedback regulation by miR168 (**Figure 7A**; **Figure 7—figure supplement 1A,B**). However, AGO1 protein retained less miRNA in the dominant negative mutant, indicating that RICEs promote integrity of functional RISC. These results were reproducible and a similar pattern could also be observed in *rice1 rice2* knockdown lines, although more modest relative to the dominant negative transgenic plants (**Figure 7—figure supplement 1A–C**). Interestingly, no obvious enrichment of miRNA*s was detected in AGO1 immunoprecipitates (**Figure 7A**; **Figure 7—figure supplement 1A–C**), suggesting that RICE1 might not clear miRNA*s that are embedded into RISC. On the other hand, we observed a coordinated decrease in the amount of miRNA and miRNA* in the transgenic plants ectopically expressing the dominant negative RICE1 mutants and the *rice1 rice2* knockdown mutants by qRT-PCR (**Figure 7B**; **Figure 7—figure supplement 1D**). To further examine the relative levels of miRNA and miRNA* on a global level, sRNA-seq was performed with wild type plants and the dominant negative RICE1 D52A mutants. The expression levels of ~40% of miRNAs in the dominant negative mutants were 1.5 fold lower than observed in the wild type, while only less than 5% were 1.5 fold higher than in the wild type (**Figure 7C**; **Supplementary file 1D**). Concurrently, the significantly downregulated and upregulated miRNA*s in the dominant negative mutants were ~35% and 9%, respectively (**Figure 7D**; **Supplementary file 1E**). Consistent with qRT-PCR results, there were also no significant changes in most of the miRNA to miRNA* ratios (**Figure 7E**; **Supplementary file 1F**).

Studies on siRNA-RISC assembly in *Drosophila* show that once the siRNA duplex is incorporated into AGO proteins, the passenger strand serves as the first target of RISC and is cleaved into 1–9 nt and 10–21 nt fragments by the catalytically-active AGO proteins before degradation (**Matranga et al., 2005**; **Rand et al., 2005**). If the miRNA-RISC assembly were similar to that of siRNA-RISC, RICE proteins might have some impact on accumulation of 9-nt and 12-nt miRNA*. To test this, we enriched a large amount of low-molecular weight RNA and repeated sRNA blotting using pooled oligo probes targeting numerous families of miRNA*s. Whereas we appeared to detect residual full-length miRNA*s, we were unable to observe any nicked miRNA* strands on the blots (**Figure 7—figure supplement 1E**). Next, we adopted the wheat germ system (**Tang et al., 2003**) to study in vitro reconstituted miRNA-RISC. Again, we failed to detect 9-nt and 12-nt miRNA* fragments in any of the reactions (**Figure 7—figure supplement 1F**). These observations are consistent

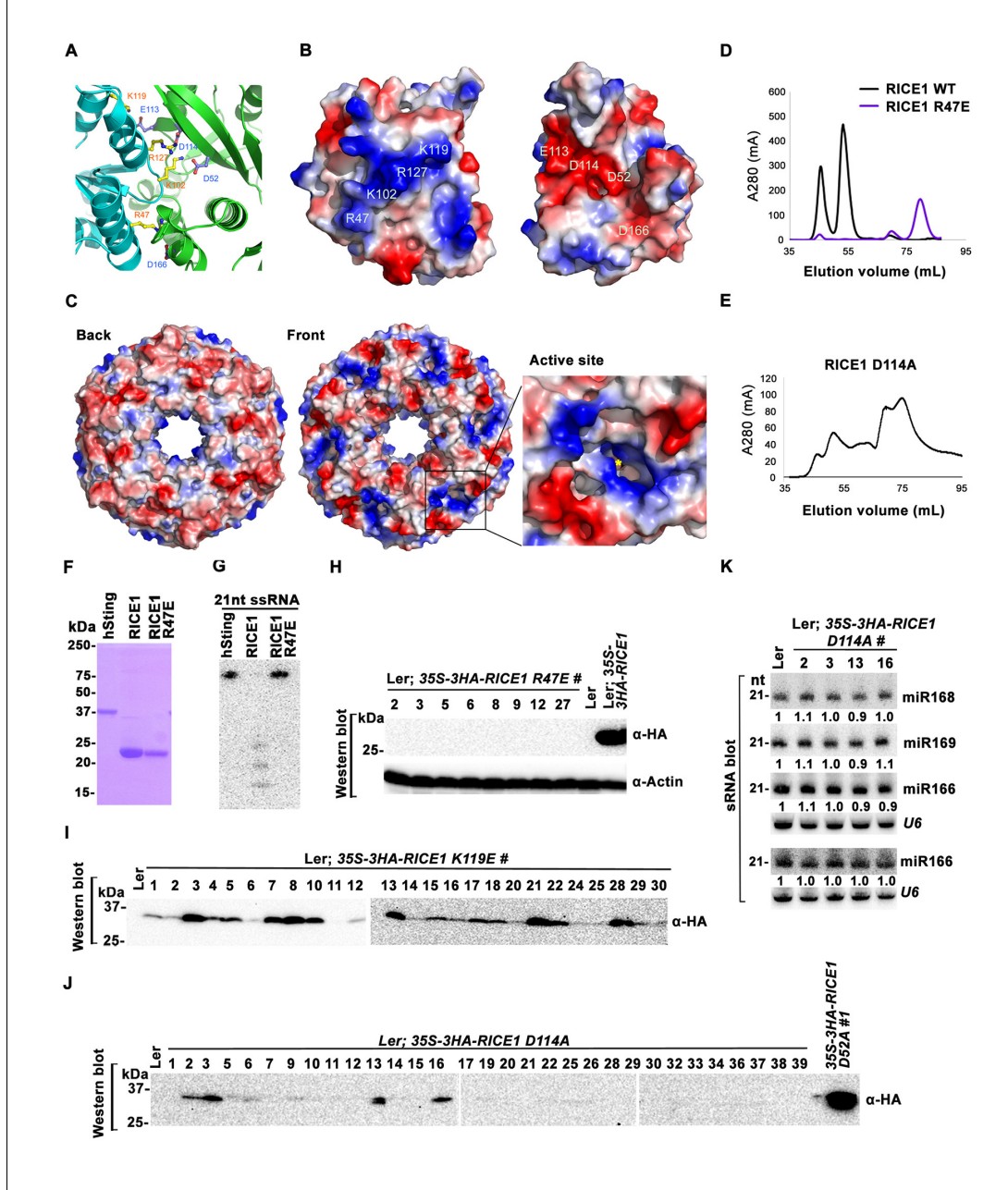

**Figure 6.** Mutations at the dimer interface impair RICE1 function and protein stability in vivo. (**A**) Interface between RICE1 monomers. Amino acid residues involved in the interaction are shown in ball-and-stick models. (**B** and **C**) The electrostatic potential at the surfaces of RICE1 monomers and hexamer. Positively and negatively charged surfaces are colored blue and red, respectively. In (**C**), the active site of RICE1 (indicated by the yellow asterisk) is located at the bottom of a deep groove between RICE1 monomers. (**D** and **E**) SEC assays of RICE1 R47E and D114A mutants defective in oligomerization. Note: SEC assay was conducted using Hiload 16/600 Superdex 200 in (**E**) instead of routinely used Superdex 75. (**F**) SDS-PAGE of purified wild type, mutated RICE1, and control protein hSting. (**G**) Oligomerization is essential for RICE1 catalytic activity. (**H**) RICE1 R47E mutant proteins failing in oligomerization are unstable in vivo. (**J**) RICE1 D114A, unlike D52A, Y54S, and K119E mutants, was unstable and barely detectable in vivo. (**K**) Ectopic expression of 3HA-RICE1 D114A had no significant effect on miRNA accumulation in vivo. Western blot analyses were performed with anti-HA antibody. sRNA blot analyses were done with the indicated $^{32}$P-labelled probes. The amount of miRNAs in wild-type plants was arbitrarily designated as 1.0, and the relative amount in the other lines was normalized to that of control plants.

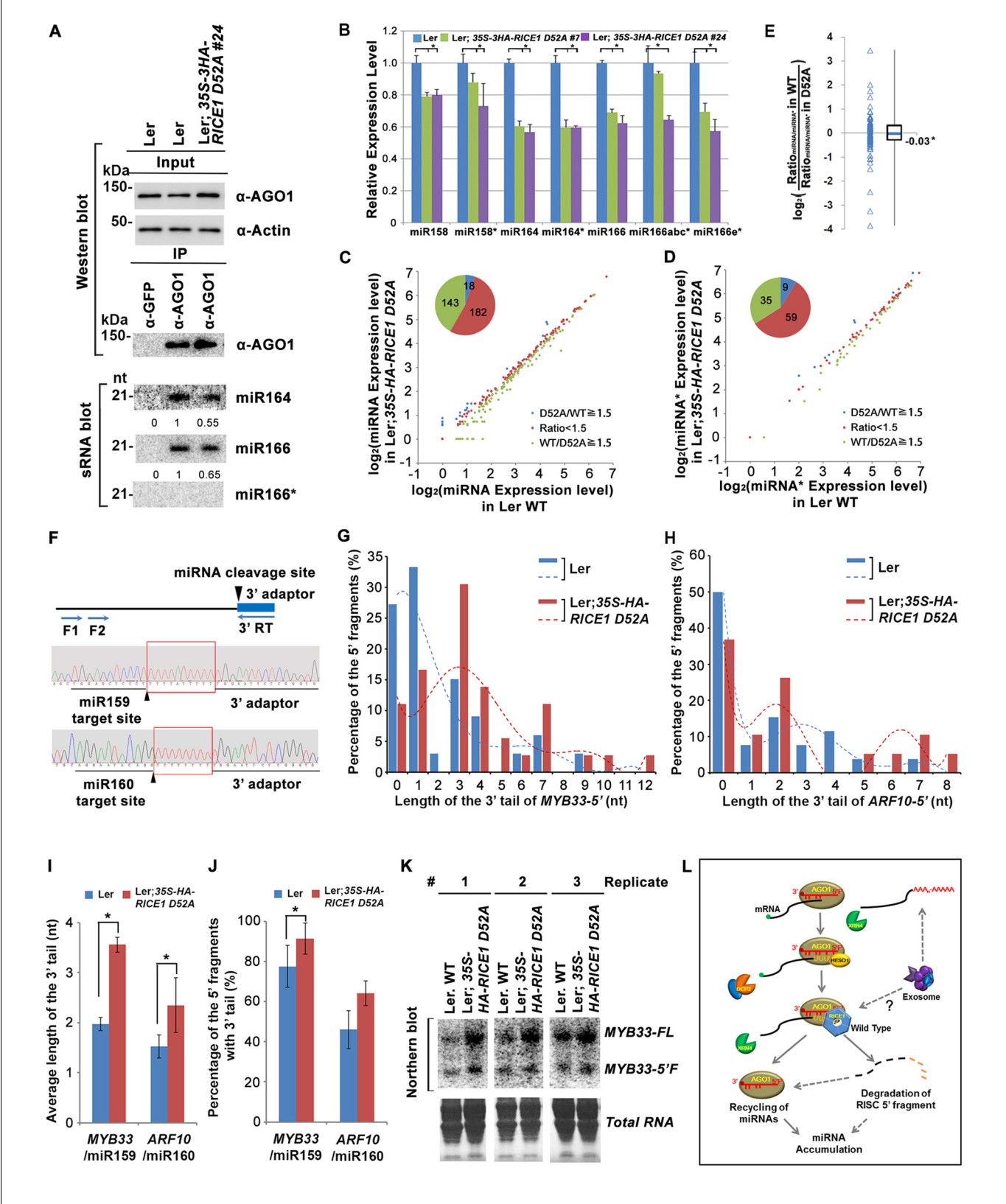

**Figure 7.** Catalytically-inactive RICE1 extends uridylated 5′ cleavage fragments from RISC activity whereas decreases the abundance of RISC-harbored miRNA in vivo. (**A**) Ectopic expression of catalytically-inactive RICE1 mutants impaired the accumulation of miRNAs in RISC. Western blot analyses were

*Figure 7 continued on next page*

*Figure 7 continued*

done with anti-AGO1 and anti-Actin antibodies. Actin serves as a loading control. sRNA blot analyses of sRNA recovered from AGO1 IP were conducted with $^{32}$P-labeled probe against miR166. The ratio of miR166/AGO1 in wild type was arbitrarily designated as 1.0, and the relative amount in each sample was normalized to that of control plants. Additional biological replicates were shown in *Figure 7—figure supplement 1A,B*. (B) Concurrent reduction of miRNA and miRNA* abundance in the transgenic plants expressing dominant-negative RICE1 mutant. The relative level of miRNA or miRNA* was normalized to that in Ler plants where the amount was arbitrarily designated as 1.0 with ±SD from at least three experiments, asterisks (*) indicate statistically significant differences (Student t-test, p<0.05). See also in *Supplementary file 1C*. (C and D) sRNA-seq analyses of miRNA or miRNA* expression in wild type and transgenic plants expressing dominant-negative RICE1 mutant. X and Y axes indicate the logarithm base two for the miRNA or miRNA* expression level in wild type and *35S-HA-RICE1 D52A* transgenic plants, respectively. The labels in the bottom right of the chart indicated the categories of miRNA or miRNA*s: D52A/WT ≥1.5, upregulated in *35S-HA-RICE1 D52A* transgenic plants for more than 1.5 fold; WT/D52A ≥1.5, downregulated in transgenic plants for more than 1.5 fold; Ratio <1.5, the rest with fold changes less than 1.5 fold. The pie in the top left of the chart indicated the numbers of different categories of miRNA or miRNA*s. See also in *Supplementary file 1D and 1E*. (E) Comparison of the ratio of miRNA vs miRNA* in wild type and transgenic plants expressing dominant-negative RICE1 mutant. The empty triangle indicates the logarithm base two for the ratios of individual pairs of Ratio$_{miRNA/miRNA*}$ in wild type relative to the ones in *35S-HA-RICE1 D52A* transgenic plants. Ratio$_{miRNA/miRNA*}$ = miRNA expression level/miRNA* expression level. *, the median value of the logarithm values mentioned above in the whole populations of tested miRNA/miRNA* pairs. See also in *Supplementary file 1F*. (F) Schematic diagram of 3' al-RACE to detect the uridine addition (red rectangle) in the 3' end of RISC-cleaved 5' fragment. The miR159- and miR160-cleavage sites were shown with black triangle. (G–I) Uridylation of 5' fragments in Ler and Ler; *35S-HA-RICE1 D52A*. The representative length distribution patterns of uridylated *MYB33-5'* (G) and *ARF10-5'* (H), and the fitting curves in dotted lines indicate the tendency of the length distribution, respectively; the mean length of the 3' tails and the mean percentage of uridylated clones in total clones were shown in (I) and (J), respectively; significant difference (Student t-test, p<0.05) was indicated with the asterisk (*). See the details for three biological replicates in *Figure 7—figure supplement 2A–D* and *Supplementary file 3A and 3B*. (K) The abundance of *MYB33-5'* in Ler and Ler; *35S-HA-RICE1 D52A*. *MYB33-5'* RNAs were detected by Northern blot in three biological replicates, using probes for *MYB33*-5' from miR159-AGO1-mediated cleavage. FL, full-length *MYB33* transcripts; 5'F, 5' fragment. Methylene blue staining of total RNAs shows the loading controls. (L) A proposed model for RICE functions in miRNA-RISC. RICEs facilitate the clearance of the uridylated tail of 5' RISC-cleaved fragments and promote RISC recycling and miRNA stability.

The following figure supplements are available for figure 7:

**Figure supplement 1.** RICEs affect the accumulation of miRNAs and miRNA*.

**Figure supplement 2.** RICEs affect the accumulation of the uridylated 5' RISC cleavage fragments.

**Figure supplement 3.** Phylogenetic analysis of RICE proteins and their orthologs in eukaryotes.

with the prevailing model of miRNA-RISC assembly in different organisms. During the loading of miRNA/miRNA* into AGO proteins, it is believed that the unwinding of the miRNA/* duplex and the removal of miRNA* are conducted through a slicer-independent mechanism due to the presence of mismatches in miRNA/* duplexes (*Kawamata and Tomari, 2010*; *Yoda et al., 2010*). Under this scenario, the central mismatches facilitate the loading of miRNA/* duplexes into AGOs, while mismatches located elsewhere promote separation of the miRNA/* duplex and the subsequent release and degradation of miRNA* (*Kawamata and Tomari, 2010*; *Yoda et al., 2010*). Similarly in plants, it is also proposed that unwinding of miRNA/* and subsequent clearance of miRNA* do not entail the cleavage activity of AGO1 (*Iki et al., 2010*; *Carbonell et al., 2012*; *Iki et al., 2012*). As such, the production of nicked 9-nt and 12-nt miRNA* would not take place; and thus it would be very unlikely that RICE proteins target the truncated miRNA*.

All together, these results further validated that RICE proteins positively regulate miRNA accumulation in vivo, while suggesting that RICEs are very unlikely to clear miRNA* to facilitate RISC assembly.

## Catalytically-inactive RICE1 leads to the accumulation of extended uridylated 5' RNA fragments generated by RISC cleavage

RISC harbors target mRNAs besides miRNA and miRNA*; and RICE1 also could cleave long unstructured ss RNA substrates. Given that unproductive RISC can decrease miRNA amount in RISC in vivo and that overexpression of *RICE D52A* reduced miRNA abundance, we next hypothesized that RICE1 might act on 5' fragments from RISC cleavage to maintain productive RISC. To test this, we conducted 3' adapter ligation-mediated rapid amplification of cDNA ends (al-RACE) assays using

total RNA prepared from Ler wild-type and *35S-HA-RICE1 D52A* lines. We tested *MYB domain protein 33* (*MYB33)* and *AUXIN RESPONSE FACTOR 10 (ARF10)* transcripts, which are the targets of miR159 and miR160, respectively, and sequenced their al-RACE products of 5' fragments from RISC activity (*MYB33-5'* and *ARF10-5'*) (*Figure 7F*). Previous studies show that 5' fragments generated by miRNA-RISC slicing undergo the 3' uridylation in Arabidopsis (*Ren et al., 2014*). Interestingly, here we found that 3' uridylation tail length of 5' fragments in *35S-HA-RICE1 D52A* lines was significantly longer than the ones in wild type control (3.56 ± 0.14 vs 1.97 ± 0.13 for *MYB33*-5' and 2.35 ± 0.55 vs 1.53 ± 0.23 for *ARF10-5'*, *Figure 7G–I*; *Supplementary file 3A and 3B*), although the percentages of uridylated *MYB33-5'* and *ARF10-5'* in Ler; *35S-HA-RICE1 D52A* is just slightly higher than the ones in Ler Wild type (91.30 ± 7.78% vs 77.61±10.48% for *MYB33*-5' and 64.03 ± 6.17% vs 45.98 ± 9.62% for *ARF10-5'*, *Figure 7J*; *Supplementary file 3A and 3B*). This pattern was reproducible in the independent biological replicates (*Figure 7—figure supplement 2A–D*; *Supplementary file 3A and 3B*). Together with the in vitro enzyme activity assay, these results suggested that RICEs are involved in the degradation of 5' fragments after uridylation modification.

In Arabidopsis, HESO1 uridylates 5' cleavage fragments and triggers their degradation. As such, 5' cleavage products exemplified by *MYB33-5'* is accumulated in a *heso1* mutant, compared with the amount in wild type (*Ren et al., 2014*). To examine how *RICE1* altered the abundance of 5' fragments, we performed RNA blot to detect *MYB33-5',* one of very few species of 5' fragments that can be readily detected in wild-type plants. We observed that the abundance of full-length *MYB33* transcript was significantly increased in *35S-HA-RICE1 D52A* transgenic line; and the accumulation was clearly due to lower amount of miRNAs in the transgenic lines. Notably, the accumulation of *MYB33-5'* which should otherwise be reduced due to less cleavage of full-length transcripts, was comparable or even slightly increased in *35S-HA-RICE1 D52A*, relative to that in Ler wild type (*Figure 7K*). Together, we propose that RICEs mediate the clearance of 5' uridylated fragments resulting from RISC cleavage and promote RISC recycling in vivo.

## Discussion

Here, using Arabidopsis AGO10 as a paradigm, we identified novel partners of miRNA-RISC, RICE1 and RICE2. We report the crystal structure of RICE1 and pivotal functions of RICEs in clearing RISC cleavage products and their regulatory role in maintaining functional RISC and miRNA abundance.

### Novel biochemical features of RICEs

Bioinformatics analysis predicted that RICEs contain an RNase H-like domain in the TAIR database. Structural analyses revealed that RICEs belong to the DnaQ-like 3' to 5' exonuclease superfamily (*Perry et al., 2006*; *Smith et al., 2013*). RICE1 also exhibits considerable structural similarity to RNase D (*Zuo et al., 2005*). Whereas RNase D is involved in processing of structural RNAs, DnaQ-like protein family, often represented by the proofreading domains of DNA polymerases, acts to check the fidelity of newly synthesized DNA during DNA replication and to excise the mismatched nucleotides in the 3'-to-5' direction. It was previously proposed that RICE2 functions as its structural homolog, WRN-exo, in DNA metabolism (*Perry et al., 2006*). However, its enzymatic activity and DNA binding capability were not previously detected (*Smith et al., 2013*). Here we demonstrate that RICEs are mainly RNases, not DNases, and function in the degradation of 5' uridylated fragments from RISC. Compared to other DEDD superfamily enzymes, RICEs exhibit some additional unique biochemical properties. First, RICEs harbor non-canonical residues in their active sites instead of the canonical DEDD motif observed in the superfamily. Moreover, RICEs lack other conserved histidine and tyrosine residues that are critical for the catalytic activity of the DEDD superfamily (*Zuo and Deutscher, 2001*). Second, the enzymatic activity of RICEs do not appear to involve metal ions, which are otherwise characteristic of DEDD superfamily proteins such as human WRN-exo (*Perry et al., 2006*), bacterial RNaseT (*Zuo et al., 2007*), yeast Ngl3p (*Feddersen et al., 2012*), among others (*Zuo and Deutscher, 2001*). Third, the active sites of RICEs are located at the interface between RICE1 subunits in the hexameric assembly; therefore, each subunit contributes to the catalytic activity. Together, the lack of a canonical catalytic motif and metal ions in the active site as well as the unique location of its active site suggest a novel catalytic mechanism in RICEs.

The multimeric assembly of RICE complex with the active sites at the interfaces between monomers in the hexameric ring may indicate physiological significance. First, the hexameric assembly of

RICEs would allow them to trap nucleic acids more efficiently for processing. Examination of surface electrostatic potential of RICE1 and RICE2 reveals that the backside of the RICE hexamers is mostly negatively charged, whereas the front surface of the hexamer contains unique positively charged patches around the active site (*Figure 6C*). Thus, these positively charged patches could sequester nucleic acids and facilitate interactions between the enzyme and substrate. On the other hand, the negatively charged surfaces might contribute to determining the orientation or polarity of 3' tails of 5' fragments that are dislodged from RISC. Second, the assembly and disassembly of RICE hexameric rings might be a dynamic process, and likely represent a new layer of homeostatic regulation of RISC maturation. A previous study showed the in vitro monomer-to-hexamer transition of RICE2 using recombinant proteins prepared from *E. coli* (*Smith et al., 2013*). In our studies, although we barely detected the presence of RICE1 monomers in solution, we did frequently observe both hexameric and dimeric conformations. Consistent with this observation, SEC profiling of plant cell extracts revealed that RICE proteins also displayed an array of monomer/dimer, hexamer, and even higher-order oligomers (*Figure 1G*). Given that the RICE monomer is very unstable, oligomerization likely allows for proper folding and protects it from protease degradation (*Figure 4*; *Figure 6H*) so the dynamic assembly and disassembly of RICE oligomers may serve as an extra regulatory layer, inferring a plastic capability of substrate processing corresponding to various functional needs.

## RICEs maintain productive RISC through clearing 5' uridylated fragments from RISC activity

Based on our biochemical and genetic results, we wished to propose that RICEs act to clear 5'uridylated fragments to maintain productive RISC and facilitate its recycling. Several pieces of evidence supported the notion: (1) RICEs directly reside on AGO proteins in cytoplasm and thus should function coordinately with RISC to regulate post transcriptional silencing events (*Figure 1*; *Figure 1—figure supplement 1*); (2) RICEs are 3'-to-5' exoribonucleases specifically targeting ss RNAs (*Figure 2*); (3) overexpression of *RICEs* elevated miRNA levels while downregulation of *RICEs* and dominant negative variants of RICE1 decreased abundance of miRNAs and their retaining in AGO proteins, indicating that RICEs are very unlikely to degrade miRNAs in vivo (*Figure 3*; *Figure 5*; *Figure 7*); (4), miRNA* was not enriched in RISC from *rice1 rice2* mutants or transgenic plants ectopically expressing dominant-negative RICEs (*Figure 7A–E*). Thus, RICEs functionally differ from some other ribonucleases that are reported to be engaged in promoting RISC assembly by clearing miRNA* (*Maiti et al., 2007*; *Xiao et al., 2010*; *Xue et al., 2012*); (5) last but importantly, overexpression of catalytically-inactive forms of RICE1 not only caused extended 5' uridylated fragments from RISC activity in vivo, but also increased their abundance (*Figure 7F–K*; *Figure 7—figure supplement 2A–D*). All the evidence implicates RICE proteins in degradation of the uridylated fragments from RISC cleavage to maintain proper function of RISC.

How do RICEs coordinate with RISC to clear the cleavage fragments? Recent single-molecule studies show that AGO proteins themselves can generate a favorable environment that actively promotes the release of the cleavage products in vitro (*Jo et al., 2015*; *Salomon et al., 2015*; *Yao et al., 2015*). In this setting, the released cleavage products might be channeled to AGO-bound RICEs for further degradation. Interestingly, we observed that hexameric assembly of RICE subunits creates a central pore with a diameter approximately 21–22 Å that is rich in positively charged residues (*Figure 6C*). It has been shown that proteins with similar ring-structures such as PCNA interact with ds DNA via their central cavity, while other ring-structured proteins translocate processively and unidirectionally along ssDNA to separate the strands of double-stranded (ds) DNA aiding both in the initiation and fork progression during DNA replication, or translocate ss RNA in transcriptional termination (*Patel et al., 2011*; *Thomsen and Berger, 2009*). Given that human AGO2 releases its cleaved products by destabilizing their interaction with the guide RNA (*Salomon et al., 2015*), RICE proteins might be coupled with a yet unidentified helicase or even AGO proteins themselves to separate double-stranded miRNA/5'-cleavage targets. The separation would allow unidirectional translocation of 5' cleavage products from RISC to RICEs for degradation, whereas retaining miRNAs inside AGO proteins (*Figure 6C*). Further study on the physiological significance of hexameric assembly of RICE proteins will provide new insight into cooperation between RICEs and RISC in RNA silencing events.

## Mechanism for RICE promotion of miRNA accumulation

Besides clearance of uridylated 5' cleavage products from RISC, RICE proteins also promote miRNA accumulation. How could RICEs enhance miRNA abundance in vivo? First, we have clarified that RICEs, different from QIP, did not impact miRNA biogenesis because the abundance of pri-miRNAs was not affected in overexpression lines of wild-type RICEs nor dominant-negative RICEs (*Figure 7—figure supplement 2E*). Second, RICE promotion of miRNA abundance is not through changing the level of the housing protein AGO1 because AGO1 amount remained stable or even slightly increased whereas miRNAs were down-regulated in the RICE-compromised lines (*Figure 7—figure supplement 1A–C*). Third, RICE proteins appeared not to target miRNA* and it is not likely that RICE proteins promote miRNA accumulation through degrading miRNA* to facilitate RISC assembly (*Figure 7A–E*; *Figure 7—figure supplement 1D–F*).

We speculate that the two functions of RICE proteins might be related. RISC performs multiple rounds of target cleavage (*Wee et al., 2012*). Prompt removal of the cleaved products promotes RISC recycling and maintains productive AGO-miRNA complexes. On the other hand, 5' cleavage fragments without prompt processing may be retained in RISC; this would result in a non-productive state that would be coupled to miRNA turnover. In this model, RICE function increases miRNA stability by clearing RISC products and promoting active RISC complexes. In the absence of RICEs, an increase in the proportion of non-productive RISC complexes would lead to miRNA turnover and thereby reduced miRNA levels. This scenario would be reminiscent of target mimicry in Arabidopsis in which, inactive interaction of targets with RISC causes miRNA decoy and repression (*Franco-Zorrilla et al., 2007*). In other organisms, complementary targets could often switch their roles from regulated substrates to regulatory RNAs to inhibit the activity of miRNAs (*Ameres et al., 2010*; *De et al., 2013*; *Xie et al., 2012*) or additional species of small RNAs (*Figueroa-Bossi et al., 2009*). Although the functions of RICE proteins in the promotion of miRNA accumulation and clearance of uridylated 5' RISC products could be interconnected, it is also possible that these two roles for RICE proteins might be independent processes. That said, the mechanism of how RICE proteins enhance miRNA expression awaits future clarification.

In conclusion, we have identified the enzymes that are specifically involved in RNA decay of uridylated 5' cleavage products from RISC. By clearing the RISC products, RICE proteins can facilitate RISC recycling and maintain an active RISC pool (*Figure 7L*). This study also raised several outstanding questions: First, how could RICEs specifically select 5' cleavage targets for degradation rather than promiscuously targeting miRNAs and miRNA* in RISC? Second, does RICEs only chop out the single-stranded part of 5' uridylated fragments, or degrade the entire 5' cleavage fragments? Considering that RICEs disfavor structured long ss RNAs, RICEs might only clear the linear ss RNA harbored in or near RISC. Until now, several candidates have been implicated in degrading the 5' cleavage products (*Branscheid et al., 2015*; *Ren et al., 2014*). In this scenario, it might be necessary for RICEs to coordinate with additional 3'-to-5' exosome nucleases or even with 5'-to-3' exonuclease and accomplish the complete clearance of RISC cleavage products. Furthermore, it has been reported that HESO1, the enzyme that accounts for uridylation of the cleavage targets, and the uridylated products are associated with AGO proteins (*Ren et al., 2014*). Thus, it is tempting to hypothesize that uridylation, release and decay of 5' cleavage fragments could be coordinated in vivo. Whether HESO1 and RICEs interacts or synergistically promote the functions for each other would be an exciting question to be addressed in the future. Equally interesting is whether there is interplay between RICEs and decapping machinery or deadenylation complexes in RNA decay. Last but not least, there are between one and four orthologs in most other angiosperm genomes (*Figure 7—figure supplement 3*); and the RICE orthologs are also conserved through animals, implying their functional significance in these organisms. Further characterization of the RICE orthologs would certainly provide new mechanistic insights into RISC functionality in the eukaryotes.

## Materials and methods

### AGO10 complex isolation

*A. thaliana* suspension MM1 cell lines expressing *Flag-4Myc-AGO10* under the control of Cauliflower Mosaic Virus 35S promoter were generated following the standard transformation procedure (*Forreiter et al., 1997*; *Menges and Murray, 2002*). Transgenic MM1 cells were collected and

ground in liquid nitrogen and AGO10 protein complexes were extracted using four volumes of extraction buffer (20 mM Tris- HCl, pH 7.5, 150 mM NaCl, 2 mM MgCl$_2$, 1 mM DTT, 1 mM EDTA, 10 μM MG132, and EDTA-free protease inhibitor cocktail [Roche]). After removal of insoluble materials by centrifugation twice at 16,000 g for 10 min at 4°C, extracts were incubated with anti-Flag M2 magnetic beads (Cat# M8823, Sigma. AB_2637089) for 2 hr at 4°C. The beads were washed five times with the extraction buffer (5 min/each time) before elution with 1.2 ml of elution buffer (extraction buffer containing 100 μg/ml 3XFlag peptide). The eluate was then incubated with anti-Myc agarose beads (Cat# A7470, Sigma. RRID:AB_10109522) for 2 hr at 4°C. The beads were washed three times with the extraction buffer, and the dual-tagged AGO complexes were eluted by incubation with 400 μl of basic buffer (0.5 mM EDTA and 30 mM NH$_4$OH) for 20 min at room temperature. The eluted immunoprecipitates were frozen in liquid N$_2$ and vacuum dried before being resolved in 4–20% gradient SDS-PAGE and proteomic analysis as described (*Castillo-González et al., 2015*). Three independent protein samples were sequenced at three different mass spectrometry facilities (Texas A and M; Cornell and Harvard Universities).

## Plant materials and growth conditions

Binary vectors including *pBA-RICE1*, *pBA-RICE1-3HA*, *pBA-RICE2-3HA*, and *pBA-3HA-RICE1* were transformed into two ecotypes (Ler and Col-0) of *A. thaliana* as well as *ago10$^{pnh-2}$* mutant background (*Lynn et al., 1999*) by the floral dip transformation method (*Zhang et al., 2006a*). The tagged or untagged RICEs were detected in seven-day-old T2 transgenic seedlings by western blot analysis. Similarly, The binary vectors harboring 35S-driven artificial miRNAs targeting specifically to *RICE1*, or *RICE2* or both were introduced into Ler background. Transgenic plants were screened for the presence of artificial miRNAs and absence of target transcripts in seven-day-old T2 transgenic plants using sRNA or northern blot analyses. Constructs of $P_{RICE1}$-*GUS* and $P_{RICE2}$-*GUS* were transformed into Col-0 background for examination of the expression profiles throughout plant growth and development.

Seeds from infiltrated plants were selected on standard MS medium containing the appropriate selective agents: 10 mg/L glufosinate ammonium (Sigma , St Louis, MO) together with 100 mg/L carbenicillin (Sigma, St Louis, MO). Homozygous progeny of T2 transgenic plants that displayed 3:1 ratio on selection plates were further identified by the lack of segregation in the presence of selective agents on MS plates.

Plants were grown in LP5 Metromix (SunGro, Canada) in a walk-in growth chamber at 22°C with relative humidity of 50% under long-day conditions (16 hr light/8 hr dark) unless otherwise noted. Light illumination (150 mmol photons/m$^2$s) was provided by a Spec-tralux T5 high-output lamp (Hydro warehouse).

## DNA construction

The majority of constructs were generated through a gateway system (*Zhang et al., 2005*). Numerous destination vectors (DC in our nomenclature) used for transient expression in *N. benthamiana* and stable Arabidopsis transformation as well as protein expression in *E.coli* were described (*Zhang et al., 2006b*; *Zhu et al., 2011*). These destination vectors include the binary gateway vectors *pBA-DC*, *pBA-Flag-4Myc-DC*, *pBA-3HA-DC*, *pBA-DC-3HA*, *pBA-3Flag-DC*, *pHyg-DC-CFP* and *pHyg-YFP-DC* as well as *E.coli*-compatible *pMAL-DC* and *pET28-SUMO-DC*. Most cDNA and artificial miRNA genes were cloned into *pENTR/D* vectors using primers listed in *Supplementary file 1G*, confirmed by sequencing, and transferred to the appropriate vectors by recombination using the LR Clonase (Invitrogen).

Promoters for *RICE1* and *RICE2* members were cloned from Arabidopsis (Col-0) genomic DNA into TOP-PCR (Invitrogen) and confirmed by sequencing. The promoter sequences covered 1184 and 1538 bp from the start codon of *RICE1* and *RICE2*, respectively. The confirmed promoter fragments were further introduced into appropriately digested binary vectors of *pBA002a-GUS* (*Zhang et al., 2005*).

*pET28-SUMO-RICE1* and -*RICE2*, which were used for protein purification, were cloned as follows: full-length cDNAs of *RICE1* and *RICE2* were cloned by PCR using two pairs of primers listed in *Supplementary file 1G*. The resultant PCR products were double digested with *BamHI* and *SacI* and

ligated into *BamHI /SacI*-treated *pET28a-SUMO* (*Lu et al., 2010*) to produce *pET28-SUMO-RICE1* and *-RICE2*. The constructs were then confirmed by DNA sequencing.

For analyses of residues potentially involved in catalytic activity and dimerization, numerous point mutations were introduced into *RICE1* using *pET28a-SUMO-RICE1* as a template and PCR-amplified using primers summarized in *Supplementary file 1G*. The constructs were sequencing confirmed before protein preparation.

## Protein expression and purification

MBP-tagged full-length or truncated AGO proteins were purified as described (*Zhang et al., 2006b*). For purification of His-tagged recombinant proteins in *E. coli*, *pET28a-SUMO-RICE1*, and *-RICE2* as well as *pET28a-SUMO-RICE1* variants were transformed into BL21 competent cells (DE3). The BL21 cells were grown at 37°C in LB medium in the presence of antibiotics (100 µg/ml kanamycin) to an $OD_{600}$ of 0.6–0.8. Protein expression was induced by addition of isopropyl-$\beta$-D-1-thiogalactopyranoside (IPTG) to a final concentration of 0.5 mM. The cells were further incubated in a shaker for overnight at 16°C before pellet collection by centrifugation at 6000 rpm for 10 min at 4°C.

For protein purification, cell pellets were re-suspended in lysis buffer (20 mM Tris-HCl pH 7.5, 150 mM NaCl, 2 mM PMSF) and disrupted by sonication at 60% amplitude for 5 to 10 min, using a Sonic Dismembrator (Model 500, Fisher Scientific). Cell lysate was cleared by centrifugation at 8000 rpm for 15 min at 4°C. Supernatant was further centrifuged at 16000 rpm for 30 min at 4°C to completely remove the cell debris. The 6His-SUMO-tagged fusion proteins were first purified by Ni-NTA resin. After the loading of soluble extracts, the resin was washed with first washing buffer (20 mM Tris-HCl pH 7.5, 500 mM NaCl, 2 mM PMSF, and 25 mM imidazole) and second washing (500 mM NaCl, 20 mM Tris-HCl pH 7.5, 2 mM PMSF, and 100 mM imidazole) buffer to remove nonspecific-binding proteins, prior to elution with elution buffer (150 mM NaCl, 20 mM Tris-HCl pH 7.5, and 250 mM imidazole).

For enzymatic assay purposes, the affinity purified RICE proteins were cleaved with SUMO protease for one hour at room temperature. The cleaved RICE proteins were further purified by gel filtration chromatography on a Superdex75 (1.6 × 60) column (GE Healthcare) eluted with the running buffer (20 mM Tris-HCl pH 7.5, 150 mM NaCl). The purity of the proteins was analyzed by a 15% sodium dodecyl sulfate (SDS) polyacrylamide gel electrophoresis (PAGE) gel, which was stained with Coomassie Brilliant blue R-250.

## In vitro pull-down assays

*RICE1* and *RICE2* open reading frames in the *pENTR* vectors were cloned to destination vector *pET28a-SUMO-DC* by LR reaction to generate His-SUMO fusion proteins. AGO10 and AGO10 truncations in the *pENTR* vectors were cloned to destination vector *pMAL-DC* by LR reaction to generate Maltose-binding protein (MBP) fusion proteins. All the plasmids were transformed into BL21 competent cells (DE3).

One microgram MBP-fusion Prey proteins of AGO1, AGO10 and AGO10 truncated proteins were preabsorbed for 1 hr at room temperature in 1 mL of binding buffer (50 µl Ni-NTA resin beads, 50 mM Tris at pH 7.5, 150 mM NaCl, 0.2% glycerol, 0.6% Triton X-100, 0.5 mM $\beta$-mercaptoethanol, 1 mM PMSF). The mixture was cleared by centrifugation at 12,000g for 2 min. The resulting supernatant was transferred to a new tube containing 1 µg of the bait protein His-SUMO/His-SUMO-RICE1/ RICE2. After incubation at room temperature for 1 hr, Ni-NTA resin beads were added, and the incubation continued for another one hour in the same conditions. Finally, six further vigorous washes were performed with the washing buffer (50 mM Tris at pH 7.5, 300 mM NaCl, 0.6% Triton X-100, 1 mM PMSF). Pulled-down proteins were resolved by 6% SDS-PAGE and detected by Western blotting using the α-MBP antibody (NEB E8032S, RRID:AB_1559732).

## In vitro transcription

Long RNA substrates of *PHV* and *At4g29770* were synthesized as described previously (*Zhang et al., 2006b*). For RNA homopolymer of 21-, 50-, 70-, and 100-nt in length, T7 promoter-fusion reverse primers listed in *Supplementary file 1G* were annealed with T7 promoter forward primer with 95°C for 10 min and naturally cooling-down in the heat block in room temperature for 1 hr; RNA homopolymers were synthesized by T7 RNA polymerase in 37°C for at least 2 hr. The RNA

transcripts were separated by 6–15% PAGE and gel-purified. The RNA transcripts were dissolved in DEPC-treated water and ready for further labeling experiments.

## RICE1 and RICE2 enzymatic assay

The RNA substrates were commercially synthesized by IDT and 5'- labeled with $\gamma$-$^{32}$P-ATP by T4 PNK. For 3' end labeling, RNA was ligated to $\alpha$-$^{32}$pCp by T4 RNA ligase followed by CIP treatment. For circular RNA substrate preparation, 5' $^{32}$P – labeled ss RNA was circularized by T4 RNA ligase I according to the manufacture's instruction. After the reaction, RNAs were extracted with a standard phenol-chloroform extraction method. The resulting RNA fragments were separated by 20% denaturing polyacrylamide gels and recovered by ethanol precipitation for overnight at −20°C.

For enzymatic assays, 0.5 μM of wild type RICEs or RICE1 variants and 5 nM of substrates were incubated for normally one hour unless specifically indicated time at 37°C in 20 μl reaction buffer containing 20 mM HEPES, pH 7.0, 50 mM KCl, 5 mM MgCl$_2$,1/10 V/V RNase inhibitor SUPERaseln (Ambion) and 1 mM DTT. The reaction mixtures were then separated with phenol-chloroform. The RNA in the aqueous phase was further precipitated with three volumes of absolute ethanol overnight at −20°C before size-fractionation on a denaturing polyacrylamide gel.

## Data collection and determination of RICE1 crystal structure

Purified RICE1 was concentrated to 20 mg/ml and crystallized by hanging drop vapor diffusion method at 16°C with ~30% pentaerylthritol proxylate (5/4 PO/OH) plus 0.2 M KCl in 50 mM HEPES at pH 7.5. The crystals were harvested in mother liquor containing 35% pentarythritol proxylate and flash frozen in liquid nitrogen. The diffraction data were collected at beamline 5.0.1 of the Advanced Light Source (ALS). The diffraction data were processed with the HKL2000 package.

The structure of RICE1 was determined by molecular replacement using MOLREP in the CCP4 package. The structural model of At5g06450 (PDB 1VK0), which has 68.5% sequence identity to RICE1, was used as search model. The model was rebuilt with Coot. The crystallographic asymmetric unit (ASU) contains three RICE1 molecules. The solvent content of the RICE1 crystal is ~80%, which corresponds to a Mathews coefficient ($V_m$) of 7.4 Å$^3$/Da. Although the crystals diffract weakly, the electron density map was clear and continuous throughout the molecules. The molecules of RICE1 in the crystal assembled into two hexameric rings, interacting with each other in a face-to-face manner. The structure was refined using the Phenix package. Statistics of data collection and structural refinement were shown in *Supplementary file 2*. All the structural figures were prepared with PyMol. Coordinates and structural factors have been deposited in the database of wwPDB with accession code 5V5F.

## Luciferase complementary imaging assay (LCI assay)

A gateway destination cassette was cloned into *pCAMBIA-NLuc* and *pCAMBIA-CLuc* (*Chen et al., 2008*) to generate the gateway destination vectors, respectively. All the test genes were cloned into the destination vectors by LR reaction. All the constructs were transformed into *A. tumefaciens* strain GV3101. LCI Assays were performed as described (*Zhang et al., 2011*). Images were acquired using an electron multiplying charge coupled device camera (EMCCD, Cascade II, Roper Scientific) and processed by WinView software (Roper Scientific).

## Confocal imaging

Agrobacterium harboring *pHyg-35S-AGO1/10-YFP* and *pHyg-35S-CFP-RICE1/2*, respectively, were infiltrated separately and/or co-infiltrated into the leaves of *Nicotina benthamiana*. The infiltrated leaves were harvested 2-days-post-infiltration and used for the confocal imaging of subcellular localization in tobacco epidermis cell by confocal microscopy (NIKON). Fluorescent signals of CFP and YFP were measured by excitation with CFPHQ Shutter (emission was detected at 485 nm) and with YFPHQ Shutter (emission was detected at 540 nm), respectively. Similarly, transgenic Col-0 seedlings expressing *pHyg-35S-CFP-RICE1/2* were collected to examine cellular localization of RICE1/2 by the confocal microscopy.

## Glucuronidase (GUS) activity assay

Histochemical analysis of the GUS activity was conducted as described (*Zhou et al., 2015*). Briefly, the germinating seeds, seedlings or other plant tissues were placed in a solution containing 0.2M Sodium Phosphate pH7.0, 2 mM $_{K3}$[Fe(CN)$_6$], 2 mM $_{K4}$[Fe(CN)$_6$], 2 mM 5-bromo-4-chloro-3-indolyl-D-glucuronide (X-Gluc), and 10% methanol) and vacuum infiltrated for about 30 min before incubation at 37°C for one or two days. The stained materials were rinsed with a series of ethanol dilutions and subsequently incubated in 70% ethanol solution overnight to remove chlorophylls. The plants were then mounted on microscope slides and examined using an Olympus DSU microscope.

## Antibody generation

Polyclonal anti-RICE1 antiserum was generated in rabbits immunized with the recombinant full-length RICE1 prepared from above-mentioned three-step purifications (Ni-NTA, SUMO protease cleavage and size-exclusive chromatography). For western blot analyses, RICE1 antibody was further affinity-purified from sera using the same recombinant proteins used for rabbit immunizations as previously described (*Zhang et al., 2005*).

## Co-immunoprecipitation experiments

For transient experiments, four-week-old *N. benthamiana* leaves were infiltrated with *A. tumefaciens* ABI harboring variety of binary plasmids. The ODs of *A. tumefaciens* cultures were diluted to 0.4. *N. benthamiana* leaves were collected two days after agroinfiltration. For *A. thaliana* plants, seven-day-old transgenic seedlings were collected from MS plates. Total protein was extracted with the immunoprecipitation (IP) buffer containing 50 mM Tris-HCl, pH 7.5, 300 mM NaCl, 4 mM MgCl 2, 5 mM DTT, 0.1% Triton-100, and the complete protease inhibitor cocktail (Roche). Cleared protein extracts were immunoprecipitated with agarose-conjugated antibodies against Flag (Sigma M8823) or Myc (Sigma A7470) as described (*Zhang et al., 2006b*).

## RNA blot and western blot analyses

Total RNA was extracted using Trizol reagent from Agrobacterium - transfected *N. benthamiana* leaves or *A. thaliana* tissues, including 7-(or 10)-day-old seedlings and three week-old adult plants depending on the desired assays or from immunoprecipitates of AGO protein complexes. RNA blot hybridizations of low-molecular-weight RNAs (sRNA blot) were performed as described (*Zhang et al., 2006b*). Each lane contained sRNAs, which were recovered from isolated AGO complexes prepared from 0.4 g tissues. For input, 10 µg of total RNA was routinely used for each sample. Blots were hybridized with $^{32}$P-radiolabeled oligonucleotide probes complementary to the sRNAs of interest. In some experiments, the same blots were stripped and re-probed with $^{32}$P-labeled oligos complementary to the indicated miRNAs. U6 served as loading controls. RNA blots were detected after exposure to a phosphor plate and quantified using the Quantity One Version 4.6.9 according to the manufacturer's instructions (Bio-Rad).

RNA blot hybridizations of high-molecular-weight RNAs (northern blot) were performed as described (*Zhang et al., 2006b*) using random-labeled probes (labeled by Amersham Rediprime II DNA labeling system, GE healthcare RPN1633) specifically targeting specific transcripts indicated.

Western blot analyses were performed as previously described using antibodies against Flag (Sigma F3165), Myc (Sigma C3956), HA (Sigma H9658), actin (A0480) as well as polyclonal antibodies against AGO1 (Agrisera) and RICE1 (generated in Rockland) (*Zhang et al., 2006b*). Western blots were developed with ECL$^+$, detected with ChemiDoc XRS+ and quantified using the ImageLab Software (Bio-Rad).

## sRNA enrichment and blot

Approximately 200 µg total RNA extracted through Trizol reagent was used each time for each sample. mRNA and rRNA were then precipitated on ice for 1 hr after addition of PEG 8000 to a final concentration of 10% and NaCl to a final concentration of 0.5M. The mixture was then centrifuged at 12,000 rpm for 10 min at 4°C. Supernatant containing sRNA was precipitated with 3 vol of ethanol at −20°C for overnight. sRNAs were pooled from several experiments before sRNA blot assays.

## In vitro miRNA/miRNA* loading assay

We used wheat germ system (*Tang et al., 2003*) to study the in vitro RISC assembly. The experimental protocol for this assay was modified from previous studies (*Iki et al., 2010*, *2012*). Flag-4Myc-AGO1, RICE1 wild type and D52A mutant protein were synthesized at 25°C for 1 hr in wheat germ extract from TNT T7 Quick Coupled transcription/translation system (Promega, L1170), mixed together (1:1, v:v) with 1 vol fresh wheat germ extract (Promega, L4380) and incubated for 30 min at 25°C in the presence of additional 0.75 mM ATP, 1.8 mM $MgCl_2$, 20 mM creatine phosphate (CP), and 0.4 mg/ml creatine kinase (CK). Then, the mixture was further mixed with 5 nM miR166/166* duplexes containing either $^{32}$P-labelled miR166 or miR166*. The reaction mixtures were collected after incubation and diluted 10-fold with TR buffer (30 mM HEPES [pH 7.4], 80 mM KOAc, 1.8 mM $MgCl_2$, 2 mM DTT, one tablet/50 ml complete protease inhibitor [Roche]) and extracted with equal volumes of TE-saturated phenol. The resulting aqueous phase was recovered, mixed with 0.1 vol of 3M NaAc (pH 5.2), 2 volumes of ethanol and 1 μL of glycogen-blue and precipitated at −20°C overnight. Spin down the pellets at 4°C in 21,000g and washed the pellets with 80% ethanol. The pellets were dried up for 5 min and dissolved in 10 μL $ddH_2O$. The dissolved RNAs were mix with an equal volume of native dye solution (1 X TBE (100 mM Tris, 90 mM boric acid, 2 mM EDTA-$Na_2$), 0,2 mg/ml bromophenol blue, 0.2 mg/ml xylene cyanol, 10% (v/v) glycerol), and analyzed with native 15–18% PAGE using 0.5 X TBE as running buffer in 4°C. The $^{32}$P signals were detected after exposure to a phosphorImager plate (Molecular Dynamics). Negative control reactions using mock-translated wheat germ were performed in parallel.

## Construction of sRNA library and data analysis

sRNAs were extracted from total RNA of whole plants of Landsberg erecta wild type and transgenic plant expressing *35S-HA-PNT1 D52A* using Trizol (Invitrogen), separated by 15% denature PAGE and sRNAs of 19–24 nt were gel-purified. sRNAs were ligated to a 3′ adaptor with/without different barcodes and a 5′ adaptor sequentially, and then RT-PCR amplified as described (*Zhu et al., 2011*; *Hafner et al., 2012*). The cDNA products were re-amplified using a pair of cloning primers and the products were gel-purified in low-melting agarose gel and subject to Illumina sequencing. After trimming adaptor sequences with fastx_toolkit-0.0.14, sRNAs with lengths between 19- to 28-nt were selected and mapped to the Arabidopsis genomic sequences (TAIR10 version) with bowtie 1.1.2 and bedtools-2.26.0. Sequences from different samples were normalized by the number of total tRNA reads with perfect genomic matches.

## Quantitative Real-Time PCR

Expression levels of miRNAs and miRNA* were examined by quantitative real-time PCR following the protocol in previous works (*Varkonyi-Gasic et al., 2007*; *Turner et al., 2013*). Total RNAs were prepared from control plants or different transformants and treated with DNase before being subjected to cDNA synthesis using Superscript III reverse transcriptase (Invitrogen) primed by miRNA/miRNA*/U6-Specific primers (*Supplementary file 1G*). Quantitative real-time RT-PCR was performed in 384-well plates with an ABI 7900HT real-time PCR system using the SYBR Green I master mix (Applied Biosystems) in a volume of 10 μl. PCR conditions included one cycle at 50°C for 2 min, one cycle at 95°C for 10 min and 50 cycles of 96°C for 10 s followed by 60°C for 1 min. The *U6* gene (*Turner et al., 2013*) was included as an internal control for normalization. Three biological replicates were performed, and the reactions were performed in triplicate for each run. The comparative $C_T$ method was used to evaluate the relative quantities of each amplified product in the samples. The threshold cycle ($C_T$) was automatically determined for each reaction by the system.

## Al-RACE

Al-RACE was performed according to the reference (*Ren et al., 2014*) with some modifications. 2 μg total RNA was first ligated to 100 pmol RNA adaptor by T4 RNA ligase. 3′ RT primer was used for first-strand cDNA synthesis. The pairs of 3′RT/MYB33 F1 and 3′RT/MYB33 F2 primers were used for nested PCRs of MYB33-5′, while the pairs of 3′RT/PHB F1 and 3′RT/PHB F2 primers were used for nested PCRs of PHB-5′. The PCR products were cloned into pGEM-T Easy Vector (Promega) and sequenced. Three biological replicates of total RNA samples were used for Al-RACE. The mean

percentages of 5' RISC fragments with 3' U tails and the mean lengths of the 3' tails were statistically calculated. Primer sequences are listed in *Supplementary file 1G*.

## Acknowledgements

We thank Drs. M Ishikawa, and M Yoshikawa for their generous sharing of plasmids and cell lines and technical support. We thank Dr. J Zhao for bioinformatics advice, Dr. L Zeng for imaging facilities, and Drs. C Kaplan and T Devarenne for careful editing of the manuscript. The work was supported by grants from NSF CAREER (MCB-1253369) and CPRIT (RP160822) to XZ.

## Additional information

### Funding

| Funder | Grant reference number | Author |
|---|---|---|
| National Science Foundation | CAREER MCB-1253369 | Xiuren Zhang |
| Cancer Prevention and Research Institute of Texas | RP160822 | Xiuren Zhang |

The authors declare that there was no funding for this work.

### Author contributions

ZZ, Data curation, Formal analysis, Validation, Investigation, Visualization, Writing—original draft, Writing—review and editing, Wrote the paper, Designed and Carried out most of genetic and molecular experiments, and part of the biochemical assays, Collected the data, Assisted with preparing the manuscript; FH, Data curation, Formal analysis, Validation, Investigation, Visualization, Writing—original draft, Writing—review and editing, Wrote the paper, Generated transgenic calli and conducted proteomics assays, Designed and conducted genetic and biochemical assays, Assisted with preparing the manuscript; MWS, Formal analysis, Investigation, Methodology, Purified RICE1 protein and solved crystal structure of RICE1 protein, Assisted with preparing the manuscript; CS, Resources, Supervision, Investigation, Methodology, Assisted with protein purification and screening conditions for RICE1 crystallization, Assisted with preparing the manuscript; CC-G, Formal analysis, Validation, Investigation, Methodology, Purified RICE1 proteins in the later stage of the project, Designed some of biochemical assays and phylogenetic analysis, Assisted with preparing the manuscript; HK, Resources, Software, Supervision, Methodology, Provided resources and methodology, Assisted with in vivo protein-protein interaction assays, Assisted with preparing the manuscript; GT, Resources, Supervision, Methodology, Provided resources and methodology, Assisted with in vitro RISC assembly assays with wheat germ system, Assisted with preparing the manuscript; MD, Resources, Software, Methodology, Writing—review and editing, Provided resources and methodology, Assisted with the confocal assays, Assisted with preparing the manuscript; PL, Resources, Data curation, Software, Formal analysis, Supervision, Methodology, Writing—original draft, Writing—review and editing, Assisted the analysis of the crystal structure of RICE1, Wrote part of the structure-related part in the paper; XZ, Conceptualization, Supervision, Funding acquisition, Writing—original draft, Project administration, Writing—review and editing, Wrote the paper, Deigned experiments, Supervised the project

### Author ORCIDs

Zhonghui Zhang, http://orcid.org/0000-0003-3732-8010
Fuqu Hu, http://orcid.org/0000-0003-2189-9724
Martin Dickman, http://orcid.org/0000-0002-6091-6921
Xiuren Zhang, http://orcid.org/0000-0001-8982-2999

## Additional files

### Supplementary files

• Supplementary file 1. Quantitative examination of miRNA and miRNA expression in wild-type and RICE compromised mutants. (A) Quantification of miRNA expression in artificial miRNA knockdown lines. (B) Quantification of miRNA expression in catalytically-inactive mutants. (C) Quantification of miRNA/miRNA* expression in catalytically-inactive mutants. (D) Expression pattern of miRNAs in wild type and 35S-HA-RICE1 D52A transgenic plants. (E) Expression pattern of miRNA* in wild type and 35S-HA-RICE1 D52A transgenic plants. (F) Ratio of miRNA/miRNA* in wild type and 35S-HA-RICE1 D52A transgenic plants. (G) Primers and oligos used in this study.

• Supplementary file 2. Statistics of crystallographic analysis for RICE1.

• Supplementary file 3. 3' RACE for 5' uridylated cleavage fragments from RISC activity in wild type and the transgenic plant overexpressing catalytic inactive RICE1. (A) Detailed sequences and ratios of 3' uridylated tails of 5' RISC cleavage fragments. (B) Statistic analysis of the average length of 3' uridylated tail (nt) and percentage of the 5' fragments with such tails.

### Major datasets

The following datasets were generated:

| Author(s) | Year | Dataset title | Dataset URL | Database, license, and accessibility information |
|---|---|---|---|---|
| Zhang Z, Zhang X, Hu F | 2017 | small RNAs in RICE mutants | https://www.ncbi.nlm.nih.gov/geo/query/acc.cgi?acc=GSE96951 | Publicly available at the NCBI Gene Expression Omnibus (accession no: GSE96951) |
| Sung MW, Li P, Zhang X | 2017 | protein structure of RICE1 | https://www.rcsb.org/pdb/explore/explore.do?structureId=5V5F | Publicly available at the RCSB Protein Data Bank (accession no. 5V5F) |

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
