## [Decision Letter]

Thank you for submitting your article "RICEs degrade uridylated cleavage fragments to maintain functional RISC in Arabidopsis" for consideration by *eLife*. Your article has been favorably evaluated by Detlef Weigel (Senior Editor) and three reviewers, one of whom is a member of our Board of Reviewing Editors. The following individual involved in review of your submission has agreed to reveal his identity: Blake C Meyers (Reviewer #2).

The reviewers have discussed the reviews with one another and the Reviewing Editor has drafted this decision to help you prepare a revised submission.

Summary:

The paper is a technical tour de force that provides firm evidence for novel 3' to 5' exoribonucleases (RICEs) involved in clearance of 5' fragments produced by Dicer cleavage of miRNA targets. The data include biochemical evidence that the RICEs interact directly with AGO1/10, that they influence the accumulation of the 5' cleavage fragments and that there is an indirect on miRNA/miRNA* accumulation. The clearance of these 5' fragments is mediated by uridylation of the 3' end of the targeted RNA by HESO1. There are also structural data showing that the ribonuclease represents a new variation on the previously characterized DNA-Q-like exonuclease with a normal DNA substrate.

Essential revisions:

1) The hypothesis that miRNA* is the RICE target is not fully tested – the assay was for miRNA* and true the data show that its level is affected in line with the effect on miRNA. However, the test should be for the accumulation of the 1-9 and 10-21 miRNA* fragments since that is what should accumulate if the RICEs are involved in clearance of these fragments. We think this point is important because the model in Figure 7 does not explain the observed effects on miRNA accumulation.

2) Given the novelty of the RICE genes, and their apparent importance to a process of fundamental importance, it would be helpful to discuss if similar numbers of genes are present in most other angiosperm genomes. In PantherDB, there are between one and three orthologs per plant genome, which may be informative at least to mention (the third one in Arabidopsis is quite distant, with almost no recognizable homology to the RICEs): http://www.pantherdb.org/panther/family.do?clsAccession=PTHR13620:SF29.

3) Subsection “RICE1 and RICE 2 act as 3’ to 5’ exo-ribonucleases specifically targeting single-stranded RNAs”, end of last paragraph: "despite previous predictions" is a bit misleading; almost everywhere we looked online at the functional annotation of the RICE genes, they were marked as " Polynucleotidyl transferase, ribonuclease H-like superfamily protein ". The first line in Araport says "Polynucleotidyl transferase, ribonuclease H-like superfamily protein; FUNCTIONS IN: 3'-5' exonuclease activity, nucleic acid binding". It thus seems like you have chosen the least correct prediction of several as a contrast. You should modify the text accordingly.

[Editors' note: further revisions were requested prior to acceptance, as described below.]

Thank you for resubmitting your work entitled "RISC-Interacting Clearing 3'- 5' Exoribonucleases (RICEs) degrade uridylated cleavage fragments to maintain functional RISC in Arabidopsis" for further consideration at *eLife*. Your revised article has been favorably evaluated by Detlef Weigel (Senior Editor) and a Reviewing Editor.

The manuscript has been improved but there are some remaining issues that need to be addressed before acceptance, as outlined below:

A) Please correct a couple of typos or clumsy wording:

Introduction, third paragraph: RISC 5' cleavage fragments are quickly degraded considering their poor detection.

Introduction, third paragraph: functions redundantly with another UTP:RNA uridylyl transferases 1 (URT1) to trigger miRNA.

Introduction, fourth paragraph: between the 3' end of a guide RNA and its target in mammalians, the majority of identified.

Subsection “RICE1 and RICE 2 act as 3’ to 5’ exo-ribonucleases specifically targeting single-stranded RNAs”, third paragraph:

whereas only a single cleavage product corresponding to one nucleotide was observed when;a 3' labeled ss RNA substrate was applied. Moreover, 3' labeled phosphate appeared to;inhibit the target degradation (Figure 2). These results indicate that RICE1 is a 3' -to-5';ribonuclease that cleaves nucleotides off the RNA substrates progressively from the 3'.

Would be better as: whereas when a 3' labeled ss RNA substrate was applied there was only a single one nucleotide cleavage product. This RNA substrate was more stable than the 5' labeled RNA and we infer that the 3' phosphate inhibited the target degradation (Figure 2). These results indicate that RICE1 is a 3' -to-5' ribonuclease that cleaves nucleotides off the RNA substrates progressively from the 3'[…]

B) There are still places where you have the selective reference to DNase related function rather than the polynucleotidyl transferase/RNaseH like family protein. Please make the revision wherever appropriate.

Subsection “RICE1 and RICE 2 act as 3’ to 5’ exo-ribonucleases specifically targeting single-stranded RNAs”, first paragraph: Members from the DnaQ exoribonuclease family, exemplified by human Warner Syndrome Helicase exonuclease (WRN-exo), are widely involved in DNA metabolism (Perry et al., 2006). A recent structural report speculates that RICE2 might function as a WRN-exo and participates in DNA degradation (Smith et al., 2013). To characterize biochemical function of RICEs, we prepared RICE proteins from *E. coli* through Ni-NTA affinity purification.

Subsection “Novel biochemical features of RICEs”, first paragraph: Bioinformatics and structural analyses revealed that RICEs belong to the DnaQ-like 3' to 5' exonuclease superfamily. This family of proteins, often represented by the proofreading domains of DNA polymerases, acts to check the

C) Thanks for the comprehensive description of why the RICEs do not clear the miRNA* cleavage products. I can see how these cleavage products would not be affected by the RICEs but I do not understand how persistence of the 5' uridylated fragment would destabilize the miRNAs or how slower recycling of RISC would reduce the abundance of the miRNA. I am not sure that any of the text in the current version of the paper really helps to explain the effect of RICEs on miRNA. Perhaps you could just leave it at that and say that there are two effects of the RICEs on the 5' uridylated fragments and on miRNAs and that they may be related?

---

## [Author Response]

*Essential revisions:*

*1) The hypothesis that miRNA* is the RICE target is not fully tested – the assay was for miRNA* and true the data show that its level is affected in line with the effect on miRNA. However, the test should be for the accumulation of the 1-9 and 10-21 miRNA* fragments since that is what should accumulate if the RICEs are involved in clearance of these fragments. We think this point is important because the model in Figure 7 does not explain the observed effects on miRNA accumulation.*

We think that the reviewers were asking us to further clarify whether RICE proteins degrade the 1-9 and 10-21 nt miRNA* fragment –the hypothetic products of miRNA* immediately after the loading of miRNA/miRNA* into RISC. We think that this is very insightful comment as it raises a fundamental issue of how miRNA/miRNA* is loaded into RISC and how miRNA* is cleared.

First of all, let us summarize the literature about RISC assembly. Our current understanding of RISC assembly, in particular, siRNA-RISC assembly, has been largely shaped by the studies in *Drosophila* (Matranga et al., 2005; Miyoshi et al., 2005; Rand et al., 2005). In this organism, once the siRNA duplex is incorporated into AGO proteins, the passenger strand serves as the first target of RISC and is cleaved by the catalytically active AGO proteins (Matranga et al. 2005; Rand et al. 2005). The nicked passenger strand (1-9 nt and 10-21 nt fragments) of siRNA is subsequently degraded by endoribonuclease or exoribunuclease. Compared tothe assembly of siRNA-RISC, the maturation process of miRNA-RISC is more complicated and also less understood. However, due to the presence of extensive mismatches in miRNA/* duplexes, the unwinding of the miRNA/* duplex and the removal of miRNA* are conducted through a slicer-independent unwinding mechanism. Under this scenario, the central mismatches facilitate the loading of miRNA/* duplexes into AGOs, while mismatches located elsewhere promote separation of the miRNA/* duplex and the subsequent release and degradation of miRNA* by a ribonuclease yet to be identified (Kawamata and Tomari 2010; Yoda et al. 2010).

Similarly in plants, it is proposed that unwinding of the siRNA duplex occurs via cleavage of the passenger strand by catalytically competent AGO1, while unwinding of miRNA/* and subsequent clearance of miRNA* do not undergo such a process (Iki et al. 2010; Carbonell et al. 2012; Iki et al. 2012).

Here in our study, our initial hypothesis was indeed that RICE proteins might degrade miRNA* to promote RISC assembly and positively regulate miRNA accumulation and we have ever tested this hypothesis for at least *two* years and also revisited it recently! In fact, our effort also included exhausted searching for 1-9^th^ and 10-21^st^ nt fragments in the RICE-compromised situations:

1) We started with in vitro assay with the nicked miRNA*s (either the 1-9^th^ or the 10-21^st^ fragments) annealed with miRNA. The scenario mimics the miRNA/miRNA* loaded into RISC. We found that unlike previously reported QIP*, Neurospora crassa* AGO-interacting 3’ to 5’ exoribonuclease (Maiti et al. 2007), RICE1 does not degrade the nicked miRNA* in vitro (Figure 2 in the manuscript).

2) We then extensively searched 21-nt miRNA* and miRNA*-derived fragments (9-nt and 12-nt) in vivo. Since we could not detect the full-length and truncated miRNA* in regular sRNA blot, we have enriched low molecular weight of sRNAs by PEG8000 through several hundred micrograms of total RNAs. We finally loaded at least 10 µg of low molecular weight RNAs for sRNA gel blot. In order to detect miRNA*s and the nicked miRNA* fragments, we pooled numerous 21-nt and truncated oligo probes complementary to miR165 (a-b)*, miR166 (a-g)*, miR159*, miR160c*, miR162b*, miR164b*, miR167b*, miR168a*, miR171a* (right panel) respectively. We appeared to detect residue amount of miRNA* in the enriched samples, but were unable to detect any miRNA*-derived fragments in RICE1 D52A mutant in lower position below 21-nt miRNA*s (new Figure 7—figure supplement 1 in the manuscript).

3) We also searched miRNA* and nicked miRNA* fragments in pooled AGO1 IP in the RICE1 D52A mutant background; and did not find any accumulation of such products, either (as shown in Figure 7 in the manuscript).

4) In plant RNAi field, an important translation system called BYL has been used to study in vitro RISC loading assays with either siRNAs or miRNAs (Iki et al., 2010, Yoda et al., 2010) We thought to adopt the BYL system described in the references (Iki et al., 2010, Yoda et al., 2010 etc.) in our research to study if RICE1 D52A mutant could promote the accumulation of the miRNA* fragments. We have worked on optimization of the system for quite a long time; but our effort was not fruitful likely due to some technical pitfalls present in our system that we could not solve. However, in the published papers with miRNA-RISC assembly in plants, 9-nt and 12-nt fragments of miRNA* were not observable (Iki et al. 2010; Iki et al. 2012), suggesting that there might not be 9 nt and 12-nt nicked miRNA* fragments at all in plants.

5) Because we never recovered 9-nt and 12-nt miRNA* fragments in RICE1 D52A mutant background in vivo and in the BYL system, we appealed to in vitro wheat germ system under the supervision of Dr. Guoliang Tang (Tang, et al., 2003) to study the RISC assembly. The experimental protocol for this assay was modified from previous studies (Iki et al. 2010; Iki et al. 2012). Flag-4Myc-AGO1, RICE1 wild type and D52A mutant protein were synthesized at 25 °C for 1hr in wheat germ extract from TNT® T7 Quick Coupled transcription/translation system (Promega, L1170), mixed together (1:1, v:v) with 1 volume fresh wheat germ extract (Promega, L4380) and incubated for 30 min at 25 °C in the presence of additional 0.75 mM ATP, 1.8 mM MgCl2, 20 mM creatine phosphate (CP), and 0.4 mg/ml creatine kinase (CK). Then, the mixture was further mixed with 5 nM miR166/166* duplexes containing either ^32^P-labelled miR166 or miR166* (indicated by red). The reaction mixtures were collected after incubation and diluted 10-fold with TR buffer (30 mM HEPES [pH 7.4], 80 mM KOAc, 1.8 mM MgCl2, 2 mM DTT, 1 tablet/50 ml complete protease inhibitor [Roche]) and extracted with equal volumes of TE-saturated phenol. The resulting aqueous phase was recovered, mixed with 0.1 volume of 3M NaAc (pH 5.2), 2 volumes of ethanol and 1μL of glycogen-blue and precipitated at -20 °C overnight. Spin down the pellets at 4 °C in 21,000g and washed the pellets with 80% ethanol. The pellets were dried up for 5 min and dissolved in 10 μL ddH_2_O. The dissolved RNA were mix with an equal volume of native dye solution (1 X TBE (100 mM Tris, 90 mM boric acid, 2 mM EDTA-Na_2_), 0,2 mg/ml bromophenol blue, 0.2 mg/ml xylene cyanol, 10% (v/v) glycerol), and analyzed with native 15-18% PAGE using 0.5 X TBE as running buffer in 4 °C. The ^32^P signals were detected after exposure to a phosphorImager plate (Molecular Dynamics). Negative control reactions using mock-translated wheat germ were performed in parallel. We have conducted this experiment for many times and two blots were shown in Figure 8 (one of them also is shown in new Figure 7—figure supplement 1) in the manuscript: the red rectangle open box was supposed to be the position for 9- to 12-nt miRNA* fragments. However, again, we never observed these truncated miR166* fragments.

Author response image 1.Effect of the D52A mutation in RICE1 on removal of miRNA*s in wheat germ.Note: (1) The residue miRNA*s in the repeat #2 might be due to incomplete annealing (that is, more molarity of miRNA* than miRNA; or dissociation of miRNA/miRNA* complexes during sample processing); (2) The red box regions are supposed to be locations for 9-nt and 12-nt miRNA* fragments if present. (3) We have done this experiment for dozens of times: results related to accumulation of miRNA and miRNA*s were not very stable, however, the failure to detect 9-nt and 12-nt was always consistent.**DOI:**
http://dx.doi.org/10.7554/eLife.24466.022

6) We have also recently used protoplast and flower-tissue to study effect of RICE proteins on the accumulation of truncated miRNA*. Again, we did not recover 9-nt and 12-nt miRNA* fragments.

In summary, we have extensively and exhaustively tested the hypothesis that RICE proteins might degrade miRNA*, as well as 9- and 12-nt miRNA* fragments, we have obtained substantial amount of negative but consistent results. Given that we have already identified the uridylated RISC-cleavage fragments as the bona fide targets for RICE proteins, we tended to believe that RICE proteins are very unlikely to clear miRNA* as well as their derived fragments. To recapitulate all of our observations, we did not propose the model that RICE proteins degrade nicked miRNA*s to promote miRNA accumulation in Figure 7.

*2) Given the novelty of the RICE genes, and their apparent importance to a process of fundamental importance, it would be helpful to discuss if similar numbers of genes are present in most other angiosperm genomes. In PantherDB, there are between one and three orthologs per plant genome, which may be informative at least to mention (the third one in Arabidopsis is quite distant, with almost no recognizable homology to the RICEs): http://www.pantherdb.org/panther/family.do?clsAccession=PTHR13620:SF29.*

Thanks for another insightful comment! Inclusion of the suggested phylogenetic information would further underscore the significance of the work and also extend the broad interest of the RICE genes. To this end, we have now added a new supplemental figure showing a phylogenetic tree for RICE genes in selected angiosperm plants and also in other eukaryotic organisms including human. We have also added related information in the Discussion part (last paragraph and Figure 7—figure supplement 3).

*3) Subsection “RICE1 and RICE 2 act as 3’ to 5’ exo-ribonucleases specifically targeting 172 single-stranded RNAs”, end of last paragraph: "despite previous predictions" is a bit misleading; almost everywhere we looked online at the functional annotation of the RICE genes, they were marked as " Polynucleotidyl transferase, ribonuclease H-like superfamily protein ". The first line in Araport says "Polynucleotidyl transferase, ribonuclease H-like superfamily protein; FUNCTIONS IN: 3'-5' exonuclease activity, nucleic acid binding". It thus seems like you have chosen the least correct prediction of several as a contrast. You should modify the text accordingly.*

Thanks for this good suggestion.

The “previous predictions” we mentioned referred to the previous report of RICE2 protein (Smith et al. 2013), in which RICE2 was predicted as a DNase, but the nuclease activity on DNA was not detected. To avoid the potential confusion, we deleted this sentence as advised. Meanwhile, we added a full annotation of RICE protein in the first paragraph of the subsection “Identification of RICEs as AGO10-bound cofactors from Arabidopsis thaliana” as “which is annotated as a 23-kDa polynucleotidyl transferase and ribonuclease H-like superfamily protein”.

[Editors' note: further revisions were requested prior to acceptance, as described below.]

*The manuscript has been improved but there are some remaining issues that need to be addressed before acceptance, as outlined below:*

*A) Please correct a couple of typos or clumsy wording:*

We appreciate very much for the editors’ careful reading of our manuscript and for pointing out our sloppiness. We (our lab members) have now carefully corrected the manuscript. Meanwhile, we have requested kind help from our colleagues (two native speakers) who have also gone through the manuscript and cleaned up additional typos and clumsy wording. English language should be fluent now.

*Introduction, third paragraph: RISC 5' cleavage fragments are quickly degraded considering their poor detection.*

Thanks. We corrected it.

Introduction, third paragraph: functions redundantly with another UTP:RNA uridylyl transferases 1 (URT1) to trigger miRNA.

Thanks. We corrected it.

Introduction, fourth paragraph: between the 3' end of a guide RNA and its target in mammalians, the majority of identified.

Thanks. We corrected it.

*Subsection “RICE1 and RICE 2 act as 3’ to 5’ exo-ribonucleases specifically targeting single-stranded RNAs”, third paragraph:*

whereas only a single cleavage product corresponding to one nucleotide was observed when;a 3' labeled ss RNA substrate was applied. Moreover, 3' labeled phosphate appeared to;inhibit the target degradation (Figure 2). These results indicate that RICE1 is a 3' -to-5';ribonuclease that cleaves nucleotides off the RNA substrates progressively from the 3'.

*Would be better as: whereas when a 3' labeled ss RNA substrate was applied there was only a single one nucleotide cleavage product. This RNA substrate was more stable than the 5' labeled RNA and we infer that the 3' phosphate inhibited the target degradation (Figure 2). These results indicate that RICE1 is a 3' -to-5' ribonuclease that cleaves nucleotides off the RNA substrates progressively from the 3'[…]*

Thank you so much for your nice editing of our manuscript. Yes, we followed this advice and corrected it.

*B) There are still places where you have the selective reference to DNase related function rather than the polynucleotidyl transferase/RNaseH like family protein. Please make the revision wherever appropriate.*

We did not mean to have the selective reference to a DNase rather than an RNase H protein. In fact, there are only two references related to RICE1 and RICE2 genes (Perry, et al., 2006; and Smith et al., 2013); and in both references RICE2 protein was predicted as a DNase. The information that RICE genes are annotated as the polynucleotidyl transferase/RNaseH like family protein is only from the TAIR website, and there is no publication for the annotation.

We also revisited the structural analysis in the DALI server and expanded the lists of the structural homologs of RICE1. The top four hits included RICE2 (r.m.s.d 0.7~1.2 Å) (Smith et al., 2013); the exonuclease domain of Werner Syndrome helicase (WRN-exo (r.m.s.d 2.4 Å to 2E61) (Perry et al., 2006); ribonuclease D (r.m.s.d 2.7 Å to 1YT3) (Zuo, et al., 2005); and the Klenow fragment of *E. coli* DNA polymerase I (r.m.s.d 3.3 Å to 2KFN). (r.m.s.d. means root mean square derivation; the smaller the number, the more the similarity). In the structural analysis, the best fit is still a DNase. However, to provide complete information, we added a new citation of RNase D in the related context in Discussion part.

Subsection “RICE1 and RICE 2 act as 3’ to 5’ exo-ribonucleases specifically targeting single-stranded RNAs”, first paragraph: Members from the DnaQ exoribonuclease family, exemplified by human Warner Syndrome Helicase exonuclease (WRN-exo), are widely involved in DNA metabolism (Perry et al., 2006). A recent structural report speculates that RICE2 might function as a WRN-exo and participates in DNA degradation (Smith et al., 2013). To characterize biochemical function of RICEs, we prepared RICE proteins from E. coli through Ni-NTA affinity purification.

We changed it as “Computational analysis predicted that RICEs might function as RNaseH proteins (from the TAIR website) or DnaQ-like nucleases (Smith et al., 2013).”

*Subsection “Novel biochemical features of RICEs”, first paragraph: Bioinformatics and structural analyses revealed that RICEs belong to the DnaQ-like 3' to 5' exonuclease superfamily. This family of proteins, often represented by the proofreading domains of DNA polymerases, acts to check the.*

See our response above. We have now added a new citation of RNase D in the related context in Discussion part.

*C) Thanks for the comprehensive description of why the RICEs do not clear the miRNA* cleavage products. I can see how these cleavage products would not be affected by the RICEs but I do not understand how persistence of the 5' uridylated fragment would destabilize the miRNAs or how slower recycling of RISC would reduce the abundance of the miRNA. I am not sure that any of the text in the current version of the paper really helps to explain the effect of RICEs on miRNA. Perhaps you could just leave it at that and say that there are two effects of the RICEs on the 5' uridylated fragments and on miRNAs and that they may be related?*

We appreciate very much for the editors’ insightful criticism on this unsolved issue – in fact, this question (Recycling of RISC and miRNA stability) is a significant but outstanding one in the RNAi field. The question has been also bothering ourselves for years and we could not address it by ourselves because of technical difficulty. That said, we followed the editors’ advice and re-wrote the related parts (in the Abstract and in the Discussion part) by saying clearly that RICEs have two functions (cleavage of 5' uridylated fragments and miRNA stability) and the two effects might be correlated.